# M⁶A reduction relieves FUS-associated ALS granules

Gaia Di Timoteo [1,8], Andrea Giuliani [1,8], Adriano Setti[1], Martina C. Biagi [2], Michela Lisi [1], Tiziana Santini[1], Alessia Grandioso[1], Davide Mariani [3], Francesco Castagnetti [3], Eleonora Perego [3], Sabrina Zappone [3], Serena Lattante[4], Mario Sabatelli[5,6], Dante Rotili[7], Giuseppe Vicidomini [3] & Irene Bozzoni [1,2,3] ✉

Amyotrophic lateral sclerosis (ALS) is a progressive neurodegenerative disease due to gradual motoneurons (MN) degeneration. Among the processes associated to ALS pathogenesis, there is the formation of cytoplasmic inclusions produced by aggregation of mutant proteins, among which the RNA binding protein FUS. Here we show that, in neuronal cells and in iPSC-derived MN expressing mutant FUS, such inclusions are significantly reduced in number and dissolve faster when the RNA m⁶A content is diminished. Interestingly, stress granules formed in ALS conditions showed a distinctive transcriptome with respect to control cells, which reverted to similar to control after m⁶A downregulation. Notably, cells expressing mutant FUS were characterized by higher m⁶A levels suggesting a possible link between m⁶A homeostasis and pathological aggregates. Finally, we show that FUS inclusions are reduced also in patient-derived fibroblasts treated with STM-2457, an inhibitor of METTL3 activity, paving the way for its possible use for counteracting aggregate formation in ALS.

Amyotrophic lateral sclerosis (ALS) is a progressive neurodegenerative disease. The gradual motoneuron degeneration caused by ALS leads to the atrophy of the innervating muscles and consequently to paralysis and death[1]. No cure for ALS exists today, with the only exception of a treatment specific for patients carrying pathogenic variants in *SOD1* gene[2].

Many cellular processes have been associated with ALS pathogenesis. Among them is the formation of aberrant stress granules (SG) due to mutant proteins, which lead to liquid-solid phase transition and protein aggregation[3]. Physiological SG are ribonucleoprotein membrane-less organelles that form to protect cells from

stress conditions. They are composed of mRNAs, RNA-binding proteins, and translation initiation factors that are sequestered in a reversible manner to regulate mRNA stability and translation under stress conditions. SGs can be induced by a variety of stressors, such as oxidative stress, hypoxia, and heat shock, and are thought to represent a key adaptive response to cellular stress[4]. Usually, SG are transient structures that disassemble when the stress is over, but they can become pathological when stress is prolonged. Defects in both SG assembly and disassembly have been linked to neurodegenerative disorders[3,5,6]. Several mutant proteins associated with ALS, such as TDP-43 and FUS, localize, together with SG markers,

¹Department of Biology and Biotechnology Charles Darwin, Sapienza University of Rome, Rome 00185, Italy. ²Center for Life Nano- & Neuro-Science@Sapienza, Fondazione Istituto Italiano di Tecnologia (IIT), Rome 00161, Italy. ³Center for Human Technologies@Istituto Italiano di Tecnologia (IIT), Genoa 16152, Italy. ⁴Section of Genomic Medicine, Department of Life Sciences and Public Health, Università Cattolica del Sacro Cuore, 00168 Rome, Italy. ⁵Section of Neurology, Department of Neuroscience, Faculty of Medicine and Surgery, Università Cattolica del Sacro Cuore, 00168 Rome, Italy. ⁶Adult NEMO Clinical Center, Unit of Neurology, Department of Aging, Neurological, Orthopedic and Head-Neck Sciences, Fondazione Policlinico Universitario A. Gemelli IRCCS, 00168 Rome, Italy. ⁷Department of Drug Chemistry and Technologies, Sapienza University of Rome, 00185 Rome, Italy. ⁸These authors contributed equally: Gaia Di Timoteo, Andrea Giuliani. ✉e-mail: irene.bozzoni@uniroma1.it

in cytoplasmic inclusions commonly found in ALS patient samples[5–7].

N6-methyladenosine (m6A) is an RNA modification known for its role in many biological processes[8]. However, very little is known about its role in the dynamics of physiological or pathological condensates.

Although m6A modification has been suggested to enhance the phase separation potential of mRNA in vitro[9,10], it has been shown to be unable to play a significant role in mRNA recruitment in SGs in vivo[11]. In this study, we investigated the interplay between SG and m6A in the context of ALS, demonstrating that while m6A plays a limited role in SG physiology in control conditions, it has a strong impact in ALS genetic contexts, such as those where the FUS protein is mutated in domains that confer to it a cytoplasmic relocation.

Indeed, using cellular models of ALS, we found that SG formed in the presence of mutant FUS differs from those of control cells not only in the number of condensates and in the rate of recovery from stress but also in RNA composition. We found that cells expressing mutant FUS were characterized by higher m6A levels and that METTL3 downregulation re-established its amount similar to control. Moreover, the recovery of normal m6A levels produced the rescue of altered SG dynamics and reverted the transcriptome composition towards the one of control cells. Noteworthy, either the overexpression of METTL3 or the downregulation of the m6A eraser ALKBH5 worsened SG properties both in control and ALS cells. Interestingly, inhibition of METTL3 activity through the inhibitor STM-2457 led to an effective reduction of FUS-containing SG formation both in neuronal cell lines as well as in fibroblasts derived from ALS patients. Finally, we observed reduced FUS confinement within condensates when m6A levels were lowered. Collectively, these data indicate the overall importance of m6A modification in the conversion of physiological SG into pathological ones and might foresee a potential therapeutic approach in those cases where protein aggregates formation has pathological implications.

## Results

### METTL3 downregulation restores the physiological RNA composition of stress granules in ALS cellular models

In order to unveil the impact of m6A on SG RNA composition in physiological and ALS-linked conditions, we used SK-N-BE cells carrying the doxycycline inducible overexpression of either wild-type FUS (FUS^WT) or mutant FUS (FUS^P525L), and constitutively expressing the SG marker G3BP1 tagged with GFP. Notably, G3BP1 condensation in SG was observed only upon stress induction. SK-N-BE is a human neuroblastoma cell line widely used to model neurodegenerative diseases[12].

Upon doxycycline induction ("Dox+"), we obtained a threefold overexpression of FUS (Fig. S1a). As expected, FUS^P525L, due to the mutation in its nuclear localization signal, is not efficiently imported in the nucleus and associates into SG upon stress induction[13] (Fig. S1b). Indeed, in this condition, its immunofluorescence signal almost completely colocalized with that of G3BP1 (Fig. S1c).

We treated FUS^WT or FUS^P525L cells with a control shRNA or with the combination of two shRNAs against METTL3, the main m6A writer acting on mRNAs, obtaining about 50% reduction of the protein (Fig. S1d and Fig. 1a). Furthermore, we evaluated the m6A reduction on mRNAs upon METTL3 decrease through the colorimetric EpiQuik assay and verified that the reduction was comparable with previously published works[14–16] (20%, Fig. S1e). Upon doxycycline treatment and oxidative stress induction with sodium arsenite, we isolated SG through immunoprecipitation with anti-GFP antibodies[17,18] and proceeded to RNA-seq (Fig. 1a and Supplementary Data 1).

SG enrichment in the immunoprecipitated fraction was confirmed by the enrichment of positive controls such as HUWE1, TRIO, and ANHAK and the lack of ATP5O mRNA used as a negative control[19] (Fig. S1f). In order to identify transcripts possibly altered in their association with SG upon m6A reduction, RNA-seq data were analyzed through "Differential Enrichment Analysis" (DEA) comparing transcripts enrichment in SG in the presence of METTL3 and upon its downregulation. We distinguished four groups of transcripts: (1) RNAs sig-

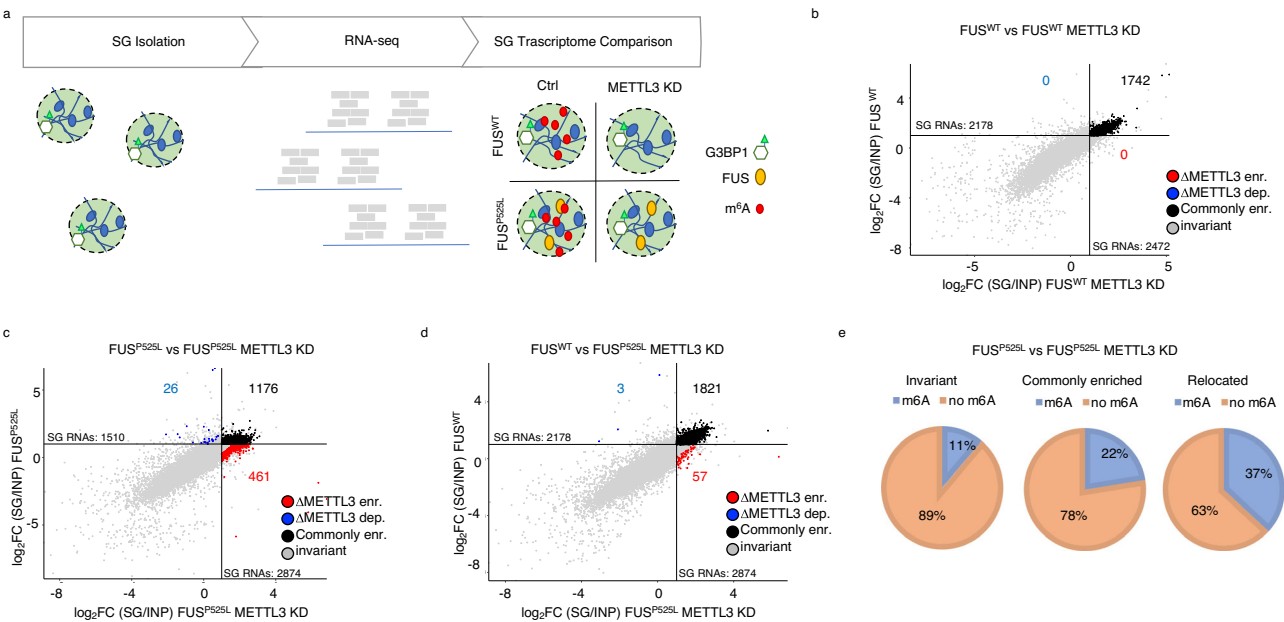

**Fig. 1 | METTL3 downregulation restores the physiological RNA composition of stress granules in ALS cellular models. a** Schematic representation of the experimental flow used for comparing stress granule transcriptomes. **b–d** Scatter plot depicting RNA differential enrichment in SG in FUS^WT vs FUS^WT METTL3-KD (**b**) or FUS^P525L vs FUS^P525L METTL3-KD (**c**), or FUS^WT vs FUS^P525L METTL3-KD (**d**) conditions. Axes describe log2FC of SG RNA enrichment in the indicated conditions. Red dots indicate *ΔMETTL3-enriched* RNAs. Blue dots indicate *ΔMETTL3-depleted* RNAs. Black dots indicate commonly enriched RNAs. Gray dots indicate invariant RNAs. *n* = 2 biologically independent replicates. **e** Pie charts displaying the percentage of meRIP-Seq enriched RNAs among invariant, commonly enriched, and relocated defined in the comparison between FUS^P525L vs FUS^P525L METTL3-KD. *n* = 2 biologically independent replicates. Source data are provided as a Source Data file.

nificantly enriched in SG both in control and METTL3 knock-down conditions, defined as "*commonly enriched*"; (2) transcripts enriched only upon METTL3 depletion, named "*ΔMETTL3-enriched*"; (3) transcripts not enriched upon METTL3 depletion, defined as "*ΔMETTL3-depleted*"; (4) RNAs not enriched or showing no significant difference in SG enrichment, defined as "*invariant*".

In agreement with previous data obtained in mES cells[11], comparing the SG-associated transcriptome of FUS^WT cells in the presence or absence of METTL3, we did not observe any differentially enriched RNA, suggesting that the reduction of m^6A does not alter SG RNA composition in wild-type conditions (Fig. 1b). No differences were also found in cells expressing only the endogenous FUS (Fig. S1g, "Dox-"), indicating that the increased levels of FUS^WT expression did not affect SG composition. In contrast, despite the overall similarity in gene expression, there were strong differences when comparing SG composition of FUS^WT *versus* FUS^P525L conditions, with 599 gained and 1267 lost transcripts in the mutant background (Fig. S1h)[18]. Remarkably, upon METTL3 downregulation, SG composition of FUS^P525L cells changed and returned very similar to that of FUS^WT relocating in SG a conspicuous fraction of RNAs (Fig. 1c, d). In fact, out of the 461 transcripts specifically recruited in FUS^P525L-containing SG, ~75% (349 out of 461) resulted in common with FUS^WT SG. For this reason, we will refer to such RNAs as "*relocated*" RNAs (Fig. S1i). This evidence was also confirmed by the differential enrichment analysis performed by directly comparing FUS^WT versus FUS^P525L in a condition of METTL3 downregulation: in fact, we observed a strong similarity between these two SG transcriptomes, with only 57 differentially enriched transcripts (Fig. 1d). Indeed, while the FUS^P525L SG transcriptome displayed strong differences with FUS^WT or Dox- control conditions, upon METTL3 downregulation the SG of wild type and mutant FUS displayed similar RNA composition (Fig. S1j). Through enrichment convergence and resampling analyses, we verified that this similarity was not dependent on the threshold adopted to define enriched RNAs in SG or on the RNA-Seq library size variability (Fig. S1k, l). Moreover, to check if the differences in SG enrichment were not due to changes in RNA expression, we compared the fold-change of RNA abundance in METTL3 knock-down and control conditions of both *relocated* and *commonly enriched* RNAs. We observed a clear increase in the fold changes of *relocated* when compared to *commonly enriched* species in the SG fraction. Such an increase was not accompanied by differences in the input samples (Fig. S1m). With the same purpose, we stratified the SK-N-BE transcriptome into five groups based on RNA expression levels (low, medium-low, medium, medium-high, and high). As displayed in Fig. S1n, both the *commonly enriched* and the *relocated* groups are mainly composed of RNAs belonging to medium expression subsets with comparable proportions, indicating that the attribution of RNAs to the *relocated* group was not due to differences in their expression levels (Fig. S1n). In agreement with these results, while the nucleotide composition of FUS^P525L SG in control conditions was significantly different[18] from all other conditions, it returned similar following METTL3 downregulation (Fig. S1o).

Overall, these data indicate that while FUS^WT and FUS^P525L SG transcriptomes are markedly different, they revert similarly under conditions of m^6A downregulation.

In order to correlate the methylation *status* of the transcripts with their localization, we performed MeRIP-Seq in FUS^P525L cells (Supplementary Data 2). The correct purification of methylated RNA was testified by the significant over-representation of the consensus DRACH motif in peak regions (Fig. S1p). Furthermore, as expected[20], the metagene plot resulting from these data displayed signal density near the stop codon (Fig. S1q). Finally, we validated by RT-qPCR the m^6A content of several RNA species identified in the MeRIP-seq, confirming the correct immunoprecipitation of m^6A-containing RNAs (Fig. S1r). Consistently, RT-qPCR results highly correlated to MeRIP-seq ones ($R = 0.82$, $p$ value = 0.024, Fig. S1s).

When analyzing the *commonly enriched* RNAs in FUS^P525L SG, we found that m^6A-containing RNAs accounted for 22% while those of *invariant* species corresponded to 11% (Fig. 1e). Such difference is mainly due to an increase in the length of the RNA (Fig. S1t), further confirming previous observations that long transcripts are enriched in SG and are more likely to contain m^6A[11,17]. Instead, the percentage of m^6A-containing RNAs increased up to 37% when looking at the *relocated* RNAs (Fig. 1e). Importantly, differently from the previous RNA subset, this enrichment is not dependent on the differential length of the transcripts since it resulted comparable to that of other transcripts included in the granules (Fig. S1t). Interestingly, the gene ontology analysis of *relocated* RNAs revealed that they mainly belong to classes related to neuron structure and function (Fig. S1u). Moreover, through a literature survey, we found that 200 over 461 genes had been previously linked to neurodegeneration or ALS (Supplementary Data 1). Interestingly, among these RNAs, we found important genes such as CLSTN1, ULK1, NGFR, and KIF5A.

The fact that, in ALS-like conditions, the downregulation of METTL3 allowed the relocation in mutant SG of a considerable fraction of RNAs, largely consisting of m^6A substrates, restoring the wild type SG RNA composition, indicated that FUS^P525L SG tend to enrich less m^6A-containing transcripts thus suggesting a link between m^6A modification and altered SG composition in ALS.

In order to investigate the open question on the relationship between FUS and m^6A modified RNAs[21,22], we performed a FUS^P525L HITS-CLIP in our experimental system (Fig. S1v and Supplementary Data 3). We found that among the direct FUS interactors (3384), only a small fraction (8%) was methylated (Fig. S1w). Moreover, peaks analysis showed that the majority of FUS binding sites do not overlap with m^6A sites (Fig. S1x) and resulted depleted for DRACH motif (Fig. S1y), indicating that m^6A target regions per se are not preferred sites for FUS binding. These data are in agreement with the observation that methylated species are less recruited in FUS^P525L SG (loss species) and only reimported upon METTL3 downregulation (*relocated* species). Overall, these results allowed us to conclude that in ALS conditions m^6A-enriched species are present at a lower percentage in SG and are not preferential targets of mutant FUS.

## METTL3 downregulation reduces the number of stress granules in ALS cellular models

In order to investigate if the downregulation of METTL3 could impact not only on the RNA content of FUS^P525L SG, but also on their number and size, we combined immunostaining for FUS and G3BP1 (Fig. 2a).

Aiming to obtain stable and homogenous METTL3 knock-down conditions, we used the CRISPR-Cas9 technology to insert a degron-tag (mAID)[23] at the N-terminus of METTL3[21] in order to obtain its auxin-inducible degradation (Fig. S2a). We obtained only heterozygous clones in both FUS^WT and FUS^P525L SK-N-BE cells (Fig. S2b). These clones showed a decrease of METTL3 both at protein (~50%, Fig. S2c, d) and RNA (~40%, Fig. S2e) levels, even without auxin induction. Notably, the downregulation of METTL3 in these conditions produced a similar decrease in its interacting partner METTL14[24] (Fig. S2c, d).

Furthermore, in order to check whether the level of METTL3 downregulation was sufficient to obtain variation in the amount of m^6A on mRNAs, we applied the EpiQuik m^6A RNA methylation quantification assay showing an approximate reduction of 20–30% (Fig. S2f). In consideration of these data and of the evidence that the expression levels of FUS^WT and FUS^P525L upon doxycycline induction was not impaired by prolonged METTL3 downregulation (Fig. S2g), we adopted the mAID-METTL3 cell lines as stable METTL3 knock-down systems for imaging studies. We confirmed by qPCR on selected mRNAs representative of the different groups that these cells, mirroring the shRNA conditions, had SG with a similar transcript enrichment (Fig. S2h). Moreover, we also performed smFISH analysis, combined with immunofluorescence for G3BP1, selecting

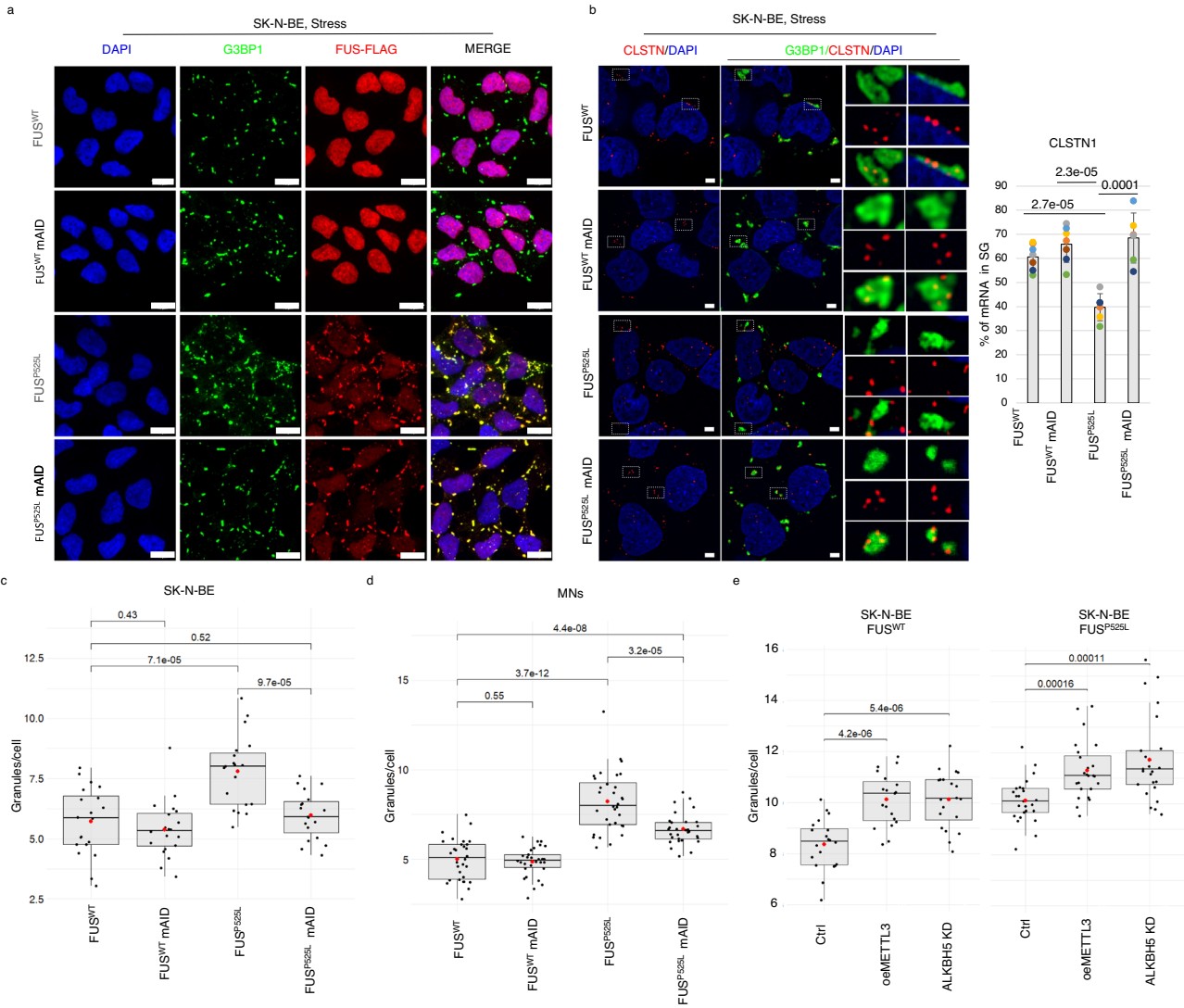

**Fig. 2 | METTL3 downregulation reduces the number of stress granules in ALS cellular models. a** Representative images of the indicated stressed SK-N-BE cells. G3BP1 antibody staining is depicted in green, FUS-Flag in red. The nuclei are stained with DAPI (blue). The merge of the signal is shown. The scale bar is 10 μm. (*n* = 3 biologically independent replicates.) **b** Representative single focal planes captions of mFISH for CLSTN1 transcript (red), immunofluorescence for stress granule marker G3BP1 (green) and DAPI (blue) in the indicated SK-N-BE cells (top). In the right panel, the digital magnification of G3BP1 granules is highlighted by dashed squares. All scale bars correspond to 3 μm. Bar plot showing representing the percentage of CLSTN1 smFISH signals colocalizing with stress granules as mean of replicates with standard deviation. Dots represent independent replicates. From 356 to #498, cells were analyzed for each condition. *n* = 3 biologically independent replicates. The *ratio* of each sample versus its control was tested by a two-tailed Student's *t*-test. *P* values are indicated. **c**, **d** Box plots illustrating the number of stress granules per cell in the indicated SK-N-BE cell lines (*n* = 3 biologically independent replicates.) or iPS-derived MN (*n* = 4 biologically independent replicates).

The box plots are defined by minima, 25% percentile, median, 75% percentile, and maximum (See Source Data file for values). The red dot indicates the average number of granules in each sample after 1 h stress. Each black dot represents the number of granules in a single field. Seven fields were acquired for each biological replicate. The *ratio* of each sample versus its control was tested by a two-tailed Student's *t*-test. *P* values are indicated. **e** Box plots illustrating the number of stress granules per cell in SK-N-BE cells either overexpressing METTL3 ("oeMETTL3") or downregulated for ALKBH5 ("ALKBH5 KD") with respect to a control condition ("Ctrl"). The box plots are defined by minima, 25% percentile, median, 75% percentile, and maximum (See Source Data file for values). The red dot indicates the average number of granules in each sample after 1 h stress. Each black dot represents the number of granules in a single field. Seven fields were acquired for each biological replicate. *n* = 3 biologically independent replicates. The ratio of each sample versus its control was tested by a two-tailed Student's *t*-test. *P* values are indicated. Source data are provided as a Source Data file.

CLSTN1 as a representative of the *relocated* group, two *invariant* species, GAPDH (not enriched in SG) and CALM1 (showing no significant difference in SG enrichment across the conditions), and HUWE1 as an example of the *commonly enriched* group. As indicated in Fig. 2b and Fig. S2i, the CLSTN1 mRNA showed a clear re-localization in SG upon METTL3 downregulation, while the other controls behave as expected, with GAPDH being always depleted from SG, CALM1 as unchanged in all conditions and HUWE1 always enriched in SG.

Through m⁶A-CLIP, we also confirmed that the global m⁶A reduction was reflected by the individual relative m⁶A level decrease for those candidates resulted methylated from the MeRIP-seq (Fig. S2j). This decrease was comparable to the one of the m⁶A-containing WTAP mRNA, not belonging to the relocated group (~30%, Fig. S2j), indicating that such group is not differentially affected by METTL3 downregulation.

We then counted the number of SG per cell upon arsenite treatment in control and in mAID-METTL3 cells expressing either FUS^WT or

FUS[P525L]. Firstly, we observed a conspicuous increase of SG in FUS[P525L] cells compared to FUS[WT] (Fig. 2c). When we analyzed mAID-METTL3 cells, we did not observe any significant variation in SG number in FUS[WT] cells; instead, we noted a great reduction in FUS[P525L] cells, where the number went back to levels comparable to those of FUS[WT] (Fig. 2c). Evaluation of SG volumes revealed no relevant average variation across the different conditions (Fig. S2k). However, when granules were divided by size into small (< 0.1 μm³), medium (0.1 μm³–1 μm³) and large (>1 μm³), we noticed that the number of large granules per cell was maintained across the samples examined, while the number of small and medium granules were responsible for their statistically significant difference (Fig. S2l).

Since ALS affects MN, we extended our analyses also to human iPSC-derived MN expressing endogenous levels of FUS[WT] or FUS[P525L][25]. In these cells we were able to obtain homozygous clones for the insertion of the degron-tag at the N-terminus of METTL3 (Fig. S2m), leading to an effective depletion of the protein, even in the absence of auxin, that was also paralleled by a strong downregulation of METTL14 (Fig. S2n). iPSCs were converted into MN according to Garone et al.[26] and analyzed after eleven days of trans-differentiation. Immuno-fluorescence experiments for METTL3 confirmed its decrease in these cells (Fig. S2o). Moreover, we also verified m⁶A RNA methylation decrease in mAID-METTL3 MN thanks to the EpiQuik quantification assay, which indicated an approximate ~30–40% reduction of mRNA m⁶A levels (Fig. S2p). The correct trans-differentiation was testified by the proper expression of three of the main MN markers (TUJ1, CHAT, and ISLE1), that was unaltered even upon METTL3 downregulation (Fig. S2q). We then compared the number and volumes of SG in FUS[WT] or FUS[P525L] MN, with or without METTL3 depletion (Fig. 2d and Fig. S2r, s). The results confirmed what was observed in SK-N-BE cells, namely that FUS[P525L] cells produce a higher number of SG which is recovered to normal levels upon METTL3 downregulation with no relevant average variation in their volumes (Fig. 2d and Fig. S2s). In this case, when

dividing SG based on their volumes, we could observe changes in both small (<0.1 μm³), medium (0.1–1 μm³), and large (>1 μm³) granules (Fig. S2t).

In conclusion, although METTL3 decrease did not affect SG number and size in a physiological context where wild type FUS is expressed, it reduced the number of SG to levels similar to the control in the FUS-associated ALS models tested. Furthermore, these data indicate that the effect of m⁶A on SG dynamics is the same in cells expressing either the endogenous or the overexpressed mutant FUS.

To deepen the link between m⁶A modification and altered SG, we compared global m⁶A levels in FUS[WT] and FUS[P525L] cells and found an m⁶A increase in ALS-associated conditions both in SK-N-BE cells and iPSC-derived MN (Fig. S2u). Furthermore, to strengthen the hypothesis of a functional link between hypermethylation and altered SG dynamics in ALS cells, we increased m⁶A levels by either over-expressing the writer METTL3 or downregulating the eraser ALKBH5 in SK-N-BE cells (Fig. S2v). Figure 2e shows that upon such treatments, both FUS[P525L] and FUS[WT] cells displayed an increased number of SG. Notably, in these conditions we did not observe FUS[WT] delocalization in the cytoplasm. These findings, alongside the observation that m⁶A decreases rescues FUS[P525L] SG number and transcriptome towards control conditions, indicate that m⁶A levels play, together with FUS, a crucial role in SG alterations in ALS.

## METTL3 decrease restores stress granules recovery rate in ALS cellular models

SG usually dissolves when stress is removed and defects in such process are linked to several human neurodegenerative diseases[27–29]. In order to investigate the effects of m⁶A on SG disassembly, we evaluated the recovery rate in METTL3 knock-down conditions through immunofluorescence analysis (Fig. 3a). The analysis of the percentage of FUS[WT] and FUS[P525L] SK-N-BE cells still containing granules after 4 h recovery from stress removal, either in control conditions or upon

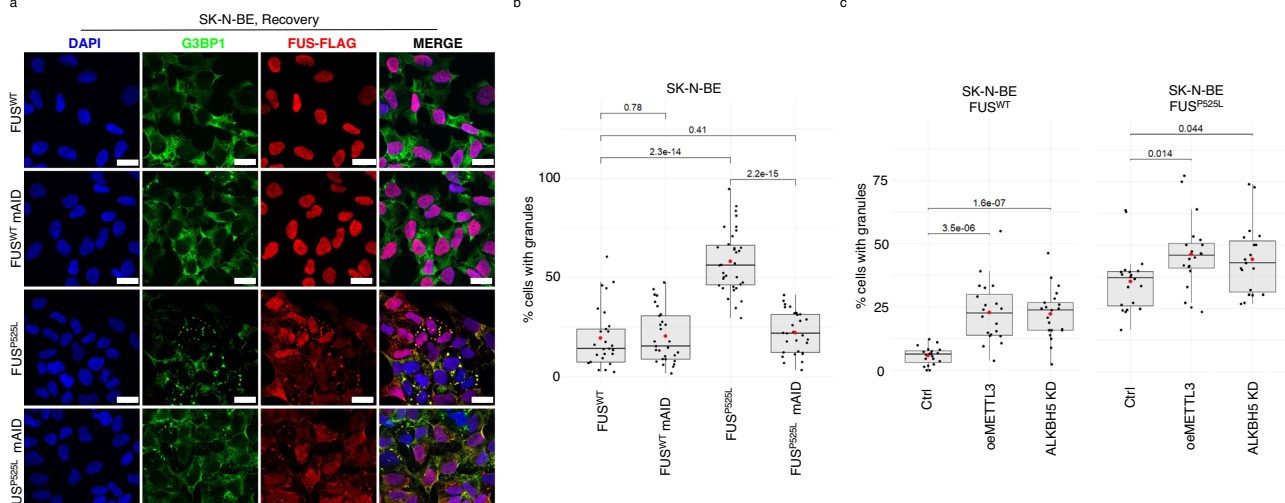

**Fig. 3 | METTL3 decrease restores SG recovery rate in ALS cellular models.**
**a** Representative images of the indicated SK-N-BE cells after 1 h stress followed by 4 h recovery. G3BP1 antibody staining is depicted in green, FUS-Flag in red. The nuclei are stained with DAPI (blue). The merge of the signal is shown. The scale bar is 20 μm. (n = 4 biologically independent replicates) **b** Box plots showing the percentage of unrecovered indicated SK-N-BE cells after 1 h stress followed by 4 h recovery. The box plots are defined by minima, 25% percentile, median, 75% percentile, and maximum (See Source Data file for values). The red dot indicates the average percentage in each sample. Each black dot represents the percentage in a single field. Seven fields were acquired for each biological replicate. n = 4 biologically independent replicates. The ratio of each sample versus its experimental

control was tested by a two-tailed Student's t-test. P values are indicated. **c** Box plots showing the percentage of unrecovered SK-N-BE cells either overexpressing METTL3 or downregulated for ALKBH5 after 1 h stress followed by 4 h recovery. The box plots are defined by minima, 25% percentile, median, 75% percentile, and maximum (See Source Data file for values). The red dot indicates the average percentage in each sample. Each black dot represents the percentage in a single field. Seven fields were acquired for each biological replicate. n = 3 biologically independent replicates. The ratio of each sample versus its experimental control was tested by a two-tailed Student's t-test. P values are indicated. Source data are provided as a Source Data file.

METTL3 downregulation, indicated that mutant cells showed a higher percentage of unrecovered cells with respect to controls (Fig. 3b). Moreover, while the depletion of the m6A writer did not change the recovery rate of FUS^WT, in mutant cells it rescued such rate to wild type levels (Fig. 3b).

Interestingly, while we observed almost total overlap of G3BP1 and FUS signals upon stress in FUS^P525L cells (Fig. S1c), we noticed granules displaying FUS-only signals after recovery (Fig. S3a). Coherently, colocalization analysis showed that the observed total overlap of the two signals under stress conditions was significantly reduced to about 40% upon recovery (Fig. S3b). Notably, the number of FUS-only aggregates decreased upon m6A downregulation (Fig. S3c), and these condensates resulted smaller than the granules observed in stress conditions (Fig. S3d). These observations suggest the hypothesis that, upon stress, SG containing both G3BP1 and FUS^P525L are formed, but after stress removal, G3BP1 is released free into the cytoplasm while FUS^P525L persists in an aggregate form. More importantly, FUS aggregation is relieved if the levels of m6A are reduced.

Consistently with the hypothesis that RNA hypermethylation can alter granules dynamics, either the overexpression of the writer METTL3 or the downregulation of the eraser ALKBH5 lowered the recovery ability both in FUS^P525L and FUS^WT cells (Fig. 3c).

## METTL3 chemical inhibition relieves FUS-containing SG

Given the availability of a small molecule, STM-2457, which has been previously validated as an effective and specific inhibitor of METTL3[30], it was worth it to verify whether it could reproduce some of the phenotypes observed in ALS cellular models.

We initially tested its activity in SK-N-BE cells expressing FUS^P525L. Figure S4a shows the effective m6A decrease upon STM-2457 treatment with respect to the DMSO control (~75% m6A reduction). The decrease of m6A levels due to STM-2457 administration recapitulated the effects of the lack of METTL3: indeed, cells formed less granules and displayed a higher recovery rate (Fig. S4b, c).

Given the availability of patient-derived fibroblasts carrying the mutant FUS^R518I allele, we applied the STM-2457 treatment to both these cells and iPSC-derived MN carrying FUS^P525L. The FUS^R518I mutation, although different from the highly pathogenic FUS^P525L, also causes re-localization of FUS in the cytoplasm, albeit to a lower extent, and causes the formation of FUS-containing SG upon stress induction (Fig. 4a). For these cellular models we assessed the m6A decrease on RNAs with the EpiQuik assay showing the efficient decrease of m6A deposition (~70–80%, Fig. S4d, e). Figure 4b, c show that the inhibition of METTL3 activity significantly reduces the number of stress granules in iPSC-derived MN as well as in patient-derived fibroblasts.

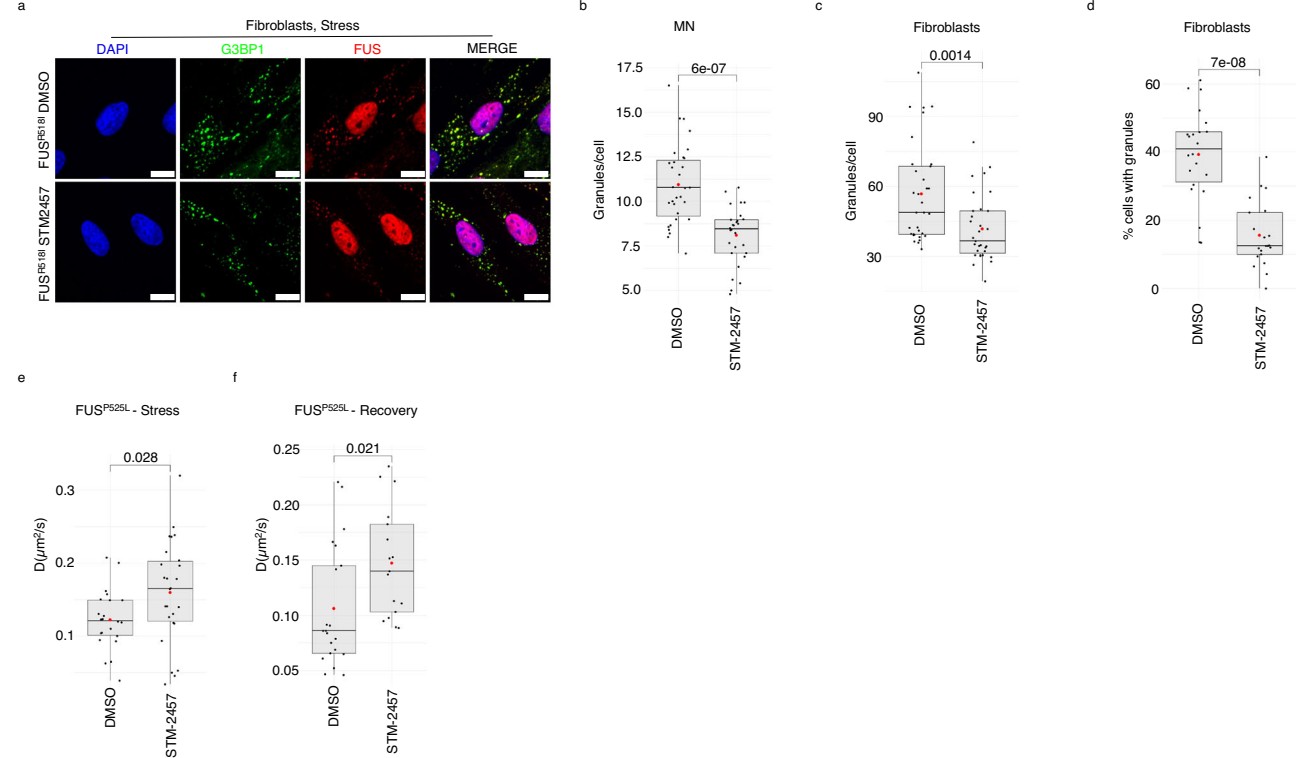

**Fig. 4 | METTL3 chemical inhibition relieves FUS-containing SG.**
**a** Representative images of the indicated stressed patients-derived fibroblasts. G3BP1 antibody staining is depicted in green and FUS in red. The nuclei are stained with DAPI (blue). The merge of the signal is shown. The scale bar is 10 μm. (*n* = 3 biologically independent replicates.) **b**, **c** Box plots illustrating the number of stress granules per cell in stressed iPS-derived MN expressing FUS^P525L or patient-derived fibroblasts treated either with DMSO or STM-2457. The box plots are defined by minima, 25% percentile, median, 75% percentile, and maximum (See Source Data file for values). The red dot indicates the average number of granules in each sample. Each black dot represents the number of granules in a single field. Seven fields were acquired for each biological replicate. *n* = 4 biologically independent replicates. The ratio of each sample *versus* its experimental control was tested by a two-tailed Student's *t*-test. *P* values are indicated. **d** Box plots showing the percentage of unrecovered patients-derived fibroblasts treated either with DMSO or STM-2457. The box plots are defined by minima, 25% percentile, median, 75%

percentile, and maximum (See Source Data file for values). The red dot indicates the average percentage in each sample. Each black dot represents the percentage in a single field. Seven fields were acquired for each biological replicate. *n* = 3 biologically independent replicates. The ratio of each sample *versus* its experimental control was tested by a two-tailed Student's *t*-test. *P* values are indicated. **e**, **f** Box plots illustrating the diffusion coefficients (D) of FUS^P525L in living SK-N-BE cells treated with either DMSO or STM-2457 in stress or recovery conditions. The box plots are defined by minima, 25% percentile, median, 75% percentile, and maximum (See Source Data file for values). The red dot indicates the average diffusion coefficient value. Each black dot is a single measurement. *n* = 2 biologically independent replicates. From 16 to 26, single measurements were carried out for each condition. The ratio of each sample versus its experimental control was tested by a two-tailed Student's *t*-test. *P* values are indicated. Source data are provided as a Source Data file.

Furthermore, in line with our previous results, we observed a lower percentage of cells with granules in patient-derived fibroblasts recovering from stress when treated with STM-2457 (Fig. 4d).

Finally, we completed the analysis of the physical properties of the SG in different contexts by using spot-variation fluorescence correlation spectroscopy (FCS) based on a single-photon array detector (see methods)[31,32]. This advanced variation of traditional FCS enables comprehensive studies of molecular mobility within living cells. In particular, we measured FUS$^{P525L}$ and G3BP1 apparent diffusion coefficient ($D$) and their confinement strength ($S_{conf}$) both under stress and recovery and either in control conditions or upon METTL3 inhibition. SK-N-BE cells expressing the GFP-tagged versions of FUS$^{P525L}$ or G3BP1 were used for such analysis (Fig. 4e, f and Fig. S4f–k). The diffusion coefficient is related to the movement of the molecules: the higher the value of $D$, the faster the diffusion rate of the molecule. The $S_{conf}$ is a metric related to the type of movement: values below 1 correlate with strong molecule confinement, while values close to 1 indicate a free diffusion behavior[32]. While the $S_{conf}$ value for G3BP1 did not change in any condition (Fig. S4h, i), we observed a lower confinement (higher $S_{conf}$ value) for FUS$^{P525L}$ in both stress and recovery when METTL3 was inhibited (Fig. S4f, g). Moreover, METTL3 inhibition produced a significant increase of the diffusion coefficient ($D$) both for G3BP1 and FUS$^{P525L}$ in stress conditions (Fig. 4e and Fig. S4j), indicating that when these proteins are present in SG assume a faster diffusion rate when m$^6$A is reduced. Furthermore, in recovery conditions while G3BP1 did not undergo D values change between control and METTL3 depletion (Fig. S4k), a significant increase of the diffusion coefficient was observed for FUS$^{P525L}$ (Fig. 4f). In addition, we found that the diffusion coefficient of G3BP1 increased from stress to recovery conditions, while that of FUS did not change (Fig. 4e–f and Fig. S4j, k), in line with the finding that while G3BP1 is released from granules in the cytoplasm, FUS$^{P525L}$ persists in an aggregate state.

In conclusion, FUS$^{P525L}$ increases its diffusion coefficient and decreases its confinement into SG when METTL3 is inhibited, further confirming the finding that reduction of m$^6$A levels relieves FUS aggregation.

## Discussion

Ribonucleoprotein (RNP) assemblies vary in a dynamic manner[6], and changes in the relative abundance of proteins and RNAs can lead to liquid-solid phase transition and eventually to the formation of aggregates[33,34]. RNP assemblies acquire significant relevance in neurodegenerative diseases where protein mutations and physicochemical insults can induce liquid-solid phase-transition[33].

ALS is associated with appearance of protein aggregates in MN[35] and several findings causatively link this phenomenon with pathological states: in fact, overexpression of either wild-type or mutant RBPs such as FUS, TDP-43 (in ALS and Frontotemporal disorders) and FMRP (in Fragile X Tremor Ataxia Syndrome) in different model organisms resulted in reduced motility and processivity of mitochondrial axonal transport and altered displacements of RNP assemblies[36–38]. Spatial localization of RNA is a very crucial process in neurons, as it controls local protein synthesis that mediates directional and morphological responses. Transport defects have been ascribed to RBPs and their ability to control correct subcellular compartmentalization of several macromolecular structures, from large organelles to mRNAs, and to regulate localized translation[39].

Therefore, the ability to prevent or revert aggregate formation could represent an important therapeutic approach for several neurodegenerative pathologies. As proof of this, molecular chaperones and pharmacological induction of autophagy have been shown to decrease FUS aggregation and alleviate associated cytotoxicity, resulting in improved neuronal survival and function[40,41]. Furthermore, antisense oligonucleotides targeting FUS have been shown to decrease FUS aggregation and rescue RNA processing defects in cellular models[42].

In this work, we demonstrate a new approach to reduce the formation of mutant FUS-containing SG and to facilitate their dissolution, consisting in reducing the levels of m$^6$A deposition on RNA. Interestingly, reduction of FUS-containing SG condensates was obtained in cell lines and in patient-derived fibroblasts also with the small molecule STM-2457, a known fully validated inhibitor of the methyltransferase activity of the METTL3-METTL14 complex.

Notably, the fact that mutant FUS showed increased diffusion and lower confinement indicated also differences in SG physical properties when m$^6$A levels were reduced. These features were paralleled by an interesting reshaping of the SG transcriptomes: while the RNA content found in FUS$^{P525L}$ was considerably different from that of FUS$^{WT}$, when m$^6$A was reduced, the transcriptome reverted towards the control one. Interestingly, we found that 50% of the transcripts lost in mutant SG and relocated upon m$^6$A reduction were previously reported to be linked to neurodegeneration and in particular to ALS. This could have important implications in the pathology since it is expected that RNA excluded from SG would not be protected from stress-related damage[43].

The deepening of this phenomenon indicated a possible molecular link between mutant FUS and m$^6$A in aberrant SG formation: we found that in FUS$^{P525L}$ cells the m$^6$A levels are higher than in control cells and lowering them rescued both the RNA content and SG dynamics to normal conditions.

Therefore, targeting this modification can indeed revert not only the morphology but also the RNA composition of these condensates. Along this line, we also show that FUS is a poor direct interactor of m$^6$A-containing RNAs. These data are in agreement with the observation that a lower percentage of m$^6$A-containing transcripts is recruited in FUS$^{P525L}$ SG (18%) in comparison to FUS$^{WT}$ (26%) and that these RNAs are relocated upon METTL3 downregulation.

Even if the specific molecular mechanism linking the presence of mutant FUS in the cytoplasm to the increase of m$^6$A cellular levels has not yet been defined, our data underline the importance of m$^6$A homeostasis, together with ALS-associated mutations, in triggering phase transition of pathological aggregates. Notably, increased m$^6$A levels were also found to be associated with another aggregate-prone ALS-linked mutation occurring in the TDP-43 protein[44]. Moreover, since pathological axonal growth was reported in multiple ALS models and m$^6$A was shown to be important for boosting such growth, m$^6$A decrease could also be beneficial for correcting this phenotype[45–47].

Finally, the finding that STM-2457 phenocopies the decrease of FUS aggregates in METTL3 knock-down cell lines opens the perspective of repurposing this molecule, previously adopted in clinical trials as an inhibitor of cell proliferation in the therapeutic treatment of acute myeloid leukemia[30], for the inhibition of protein aggregates in FUS-associated ALS cases.

## Methods

Our research complies with all relevant ethical regulations and was conducted in accordance with the criteria set by the Declaration of Helsinki.

Human primary fibroblasts were obtained from a patient carrying the p.R518I in *FUS* gene, after written informed consent and approval of the Ethic Committee (Protocol nr. A.133/CE/2013).

### Cell culture and preparation

SK-N-BE cells were cultured in RPMI (Sigma-Aldrich, Saint Louis, MO, USA) supplemented with 10% FBS (Sigma-Aldrich, #F2442), GlutaMAX supplement 1X (Thermo Fisher Scientific, # 35050061), sodium-pyruvate 1 mM (Thermo Fisher Scientific, #11360070) and Pen/Strep 1X (Sigma-Aldrich, #P4458). For the inducible expression of FUS$^{WT}$ or

FUS^P525L, SK-N-BE cells were exposed to 50 ng/mL Doxycycline (Sigma-Aldrich, #D9891) for 24 h before the sodium arsenite treatment (Sigma-Aldrich, #106277). For the immunofluorescence experiments, SK-N-BE cells were dissociated with Trypsin 1X (Sigma-Aldrich, #T4549) and 20,000–30,000 cells/well were plated on glass coverslips in 24-well plates.

Skin biopsy was performed at the distal leg using a 4-mm punch. Skin samples were dissected and cultured in BIOAMF-2 (Biological Industries) complete medium. Our research complies with all relevant ethical regulations and was conducted in accordance with the criteria set by the Declaration of Helsinki. Human primary fibroblasts were obtained from a patient carrying the p.R518I in *FUS* gene, after written informed consent and approval of the Ethic Committee (Protocol nr. A.133/CE/2013). Human primary fibroblasts were cultured in DMEM high-glucose with 1 mM sodium-pyruvate (Sigma-Aldrich, #P5280) supplemented with 20% FBS, L-glutamine 2 mM, and Pen/Strep 1X. Cells were dissociated with trypsin-EDTA 1X (Corning, #25-053-CI) and 50,000 cells/well were plated on collagen-coated glass coverslips in 24-well plates. The following day, fibroblasts were treated with 15 μM STM-2457, while control fibroblast were treated with same volumes of DMSO (Sigma-Aldrich, #D2650), for 48 h before the sodium arsenite treatment. Human NIL iPSCs were cultured in Nutristem (Sartorius, #05-100-1 A) supplemented with Pen/Strep 0.1% on geltrex-coated plates (Thermo Fisher Scientific, #A1413202) and differentiated towards spinal motoneurons as described by ref. 26. After 5 days of differentiation, cells were dissociated with Accutase (Thermo Fisher Scientific, #00-4555-56) and 200,000 cells/well were plated on geltrex-coated glass coverslips in 24-well plates. After an additional 6 days of differentiation, the iPSC-derived motoneurons were treated with sodium arsenite. As regards for METTL3-inhibition experiments, MNs were treated with 15 μM STM-2457, while control MN were treated with the same volumes of DMSO, for 48 h before the arsenite treatment. All cell lines used in were grown at 37 °C, 5% CO₂. All cell lines were tested for mycoplasma contamination. For acute oxidative stress treatment, SK-N-BE cells, iPSC-MNs, and primary fibroblasts were exposed to 0.5 mM sodium arsenite for 1 h. Rescue experiments were performed by restoring normal growth medium for 4 or 2 h after acute arsenite treatment in SK-N-BE cells or fibroblasts, respectively. For METTL3 overexpression and ALKBH5 downregulation experiments, SK-N-BE cells were plated on a 35 mm plate and transfected with lipofectamine 2000 transfection reagent (Thermo Fisher Scientific, #11668019), with either 1 μg of a vector expressing METTL3 or with the combination of 500 ng of sh1 and 500 ng of sh2 against ALKBH5 (TRCN0000291769 and TRCN0000291838), respectively. Transfection with 1 μg of a vector expressing scramble shRNA was used as a control (SHC202 TRC2). Eight hours after transfection 30,000 cells of each condition were trypsinized and transferred on glass coverslips in 24-well plates. The day after, FUS expression was induced with 50 ng/mL doxycycline for 24 h, and acute oxidative stress or recovery conditions were obtained as described below.

## Chemicals
STM-2457 has been prepared according to published procedures[30].

## Oligonucleotides
Oligonucleotides and smFISH probes used in this study are listed in Supplementary Data 4.

## Genome editing
mAID-tag was inserted at the N-terminus of METTL3 together with the coding sequence for the hygromycin resistance and a p2A signal for a proteolytic cut in SK-N-BE and iPS cells by Crispr/Cas9-induced DNA break. Two guide RNAs targeting the 5′UTR region of METTL3 were designed with CHOPCHOP and cloned in a px330 vector through BbsI digestion. The plasmid donor carrying the sequence to be inserted plus 500 nt of homology upstream and downstream was synthetized

by GENEWIZ. 10⁶ SK-N-BE cells were transfected with 3 μL of Lipofectamine 2000 in 300 μL of Optimem with donor 2 ug of DNA donor and 2 ug of each guide plasmid. The same amount of DNA was transfected in 10⁶ iPS cells with Neon™ Transfection System 100 μL Kit (Invitrogen, #MPK10096) according to the manufacturer's instruction (1 pulse, 20 V, 30 ms). Cells transfected with donor only were used as selection control. The medium was replaced with fresh one added with 450 μg/mL or 300 μg/mL hygromycin 24 h after transfection for SK-N-BE and iPS cells, respectively. Selection was carried out until control cells were died. Single colonies were transferred to 24-well plates. Colonies were then split, and a half was used for gDNA extraction and genotyping through a Rapid Extraction PCR kit (PCR biosystems, #PB10.24). Oligos 9F and 4R were used for amplifying the WT and recombinant alleles (Supplementary Data 4).

## Immunocytochemistry
Cells were fixed for 20 min at RT with cold 4% paraformaldehyde (Electron Microscopy Sciences, #15710) diluted to in complete PBS (Sigma-Aldrich, #D1283), rinsed 3 times with complete PBS and stored in PBS at 4 °C. Cells were permeabilized with 0.3% Triton X-100 (Sigma-Aldrich, #648466) diluted in complete PBS for 10 min and blocked for 30 min at RT with 5% goat and/or donkey serum, depending on the host species of the secondary antibody used. Samples were then incubated overnight at 4 °C with the primary antibody diluted in blocking solution (5% goat and/or donkey serum in PBS, #G9023 and #D9663 Sigma-Aldrich). Cells were washed with complete PBS three times for 5 min at RT and then incubated with the following secondary antibodies for 1 h at RT diluted in blocking solution and were incubated for 1 h at RT. Nuclei were stained with 1 μg/ml DAPI (#D9542, Sigma-Aldrich) diluted in complete PBS for 5 min and coverslips were mounted applying ProLong™ Glass Antifade Mountant (Thermo Fischer Scientific, #P36980) leaving the slides overnight on the bench. Confocal images were acquired with an inverted Olympus iX73 equipped with an X-Light V3 spinning disc head (Crest Optics), a Prime BSI Scientific CMOS (sCMOS) camera (Photometrics) and MetaMorph software (Molecular Devices), as Z-stacks (0.3-um step size) with a 60× oil-immersion objective.

The following antibodies were used for immunohistochemistry in this study:

anti-G3BP1 rabbit antibody (1:300, ab181150 Abcam)

anti-FUS mouse antibody (Santa Cruz Biotechnology, #sc-47711, 1:300)

anti-FLAG mouse antibody (1:400, F1804 Sigma-Aldrich)

anti-METTL3 rabbit monoclonal antibody (Abcam, #ab195352, 1:1000)

anti-Beta III Tubulin (TUJ1) chicken antibody (1:400, AB9354 Sigma-Aldrich)

goat anti-rabbit Alexa Fluor 488 (1:300, A11008 Thermo Fisher Scientific)

goat anti-mouse Alexa Fluor 488 (1:300, A11001 Thermo Fisher Scientific)

goat anti-rabbit Alexa Fluor 555 (1:300, 111-165-003 Jackson Immunoresearch)

goat anti-mouse Alexa Fluor 555 (1:300, 115-165-003 Jackson Immunoresearch)

goat anti-chicken Alexa Fluor 594 (1:300, ab150176 Abcam)

donkey anti-mouse Alexa Fluor Plus 647 (1:300, A32787 Thermo Fisher Scientific)

donkey anti-rabbit Alexa Fluor Plus 647 (1:300, A32795 Thermo Fisher Scientific)

## Immunocytochemistry image analysis
The analysis of immunofluorescence images was performed through the ImageJ software[48]. G3BP1 was used as an SG marker. The ImageJ

tool "3D Object Counter" was used for the quantification of SGs number and volumes. For the estimation of cells forming SGs in the recovery experiments cells were scored as SG-positive when they presented at least one G3BP1 spotted cytoplasmic signal. For the intensity fluorescence analysis, a nuclear mask was generated using the DAPI signal to specifically select nuclear regions; this mask was then used to measure the METTL3 fluorescence intensity mean of each nucleus with the ImageJ tool "Measure". For the signal colocalization analysis the ImageJ plugin "Jacop" was used to calculate the Manders coefficient of FUS signal colocalizing with the G3BP1 signal. In order to consider only the cytoplasmatic signal of FUS, a nuclear mask was created for each image and removed from both the FUS and G3BP1 images prior to proceed with the analysis. For the signal profile analysis random granules were selected and sectioned with a 20-pixel length straight line and the ImageJ tool "Signal Profile" was used to quantify the signal intensity. For the quantification of "FUS-only" granules in recovery experiments, the G3BP1 signal was used to create a mask and then removed from the FUS signal before the quantification of the number and volumes of FUS-only granules using the ImageJ tool "3D Object Counter".

## smFISH

FISH Probes against CALM1, HUWE1, and CLSNT1 transcripts were designed by utilizing the Stellaris® RNA FISH Probe Designer (Biosearch Technologies, see Supplementary Data 4) and labeled with ddUTP-Atto633 (AD 633-31 and ATTO-TEC) following ref. 49. FISH probes for GAPDH mRNA, labeled with Quasar 570 Dye (SMF-2026-1), were purchased from LGC Biosearch Technologies. Hybridization steps were performed according to the Stellaris RNA FISH protocol for adherent cells. Samples were then incubated overnight at 4 °C with the primary antibody for G3BP1 diluted in a blocking solution (5% goat serum in PBS, #G9023 Sigma-Aldrich). Cells were washed with complete PBS three times for 5 min at RT and then incubated with the corresponding secondary antibodies diluted 1:200 in 2% BSA/PBS for 45 min at room temperature. Nuclei were stained with 1 µg/ml DAPI (#D9542, Sigma-Aldrich) diluted in complete PBS for 5 min and coverslips were mounted applying ProLong Diamond Antifade Mountant (#P-36961, Thermo Fisher Scientific). Samples were imaged using an inverted confocal Olympus IX73 microscope equipped with a Crestoptics X-LIGHT V3 spinning disk system and a Prime BSI Express Scientific CMOS camera and with an Olympus iX83 FluoView1200 laser-scanning confocal microscope. The acquisitions were obtained using a UPLANSApo 60X (NA 1.35) oil objective and collected with the MetaMorph software (Molecular Devices). Transcript enrichment in G3BP1 granules were computed on MIP (Maximum Intensity Projection of Z-planes) of Z-stack (300 nm Z-spacing) after subtracting the FISH signals colocalizing with the nucleus. FISH spot quantification was performed by using RS-FISH[50] plugin in FiJi software over the entire microphotograph field in order to analyse from a minimum of 283 to a maximum of 650 cells from three biological replicates in each experimental condition.

## Protein analyses

Protein analyses were carried out as described in ref. 51.

The following antibodies were used in western blots for protein analyses:

Anti-FUS mouse antibody (Santa Cruz Biotechnology, #sc-47711, 1:300)

Anti-METTL3 rabbit monoclonal antibody (Abcam, #ab195352, 1:500)

Anti-METTL3 [EPR18810] monoclonal antibody (Abcam, #ab195352, 1:1000)

Anti-METTL14 polyclonal antibody (Atlas, #HPA038002, 1:1000)

Anti-Flag M2-Peroxidase (HRP) (Sigma-Aldrich, #A8592, 1:2500)

Anti-ACTB-Peroxidase (AC-15) monoclonal antibody (Sigma-Aldrich, #A3854, 1:2500)

Anti-GAPDH (6C5) monoclonal antibody (Santa Cruz Biotechnology, #sc-32233, 1:1000)

Anti-Rabbit IgG (H + L) Secondary Antibody, HRP (Thermo Fisher Scientific, #31460, 1:10,000)

Anti-Mouse IgG (H + L) Secondary Antibody, HRP (Thermo Fisher Scientific, #32430, 1:10,000)

Anti-Actinin monoclonal antibody (Santa Cruz Biotechnology, #sc-390205, 1:1000)

Anti-HPRT (HRP) antibody (Santa Cruz Biotechnology, #sc-20975, 1:1000)

Anti-ALKBH5 monoclonal antibody (Proteintech, #67811-1-Ig, 1:1000)

## RNA analyses

RNA analyses were carried out as described by ref. 52. Oligonucleotides used for this study are listed in Supplementary Data 4.

## m⁶A RNA methylation quantification

After polyA+ RNA selection through Poly(A)Purist-MAG kit (Invitrogen, #AM1922), $m^6A$ level quantification was performed thanks to the EpiQuik $m^6A$ RNA Methylation Quantification Kit (EPIGENTEK, # P-9005) according to manufacturer's instructions using 50–200 ng of RNA per well.

## Spot-variation fluorescence correlation spectroscopy measurements

To perform the spot-variation FCS measurements, we used a custom laser-scanning microscope equipped with a 5×5 single-photon avalanche diode (SPAD) array detector described by ref. 32. Cells were seeded onto a µ-Slide eight-well plate (Ibidi GmbH) and imaged in Live-Cell Imaging Solution (Thermo Fisher Scientific) at 37 °C. Before each measurement, cells were visually inspected by imaging. The axial position for the spectroscopy measurements was placed in the middle of the chosen cell. Multiple planar positions in cells were selected to probe different points. Measurements were performed within SGs in stress and in the cytoplasm in recovery conditions. The fluorescence intensity was acquired for about 100 s and analyzed offline. All measurements were performed at 37 °C inside a temperature-controlled chamber (Bold Line Temperature Controller, Okolab, PA, USA). In spot-variation FCS, conventional single-point FCS is performed for different detection volume sizes, at the same sample position. By plotting the diffusion time, as a function of the detection volume size, namely the diffusion-law, it is possible to identify the modality of mobility of the molecule[31,53,54]. Spot-variation FCS can be significantly simplified by employing a novel SPAD array detector[32]. This array detector allows for the simultaneous probing of different detection volumes, as opposed to a sequential series of measurements, thereby reducing the complexity of the measurement process and providing a clearer insight into the temporal heterogeneity in the dynamics. We calculated the time correlations directly on the lists of absolute photon times[55]. The data was then split into chunks of 10 s, and the time autocorrelations were calculated for each chunk. To calculate the apparent diffusion coefficient, we used the larger detection volume formed by all 5 × 5 elements of the detector. The lists of all SPAD channels were merged and the correlations were calculated. The individual correlation curves were visually inspected (Fig. S4l), and all curves without artifacts were averaged and fitted with a function describing a 3D diffusion. The diffusion coefficients are calculated from the fitted diffusion times knowing the confocal volumes of the microscope (calibrated as described by refs. 32,46. The confinement strength is the ratio between D measured for the smallest detection volume size (central element of the detector) and D measured for the biggest one and it is a

metric related to the type of mobility[32], which we introduced to represent the diffusion-law behavior with a single value. $S_{conf}$ values below 1 correlate with strong molecule confinement (transient binding diffusion), while values closer to 1 indicate a free diffusion behavior. In spot-variation FCS, conventional single-point FCS is performed for different detection volume sizes, at the same sample position. By plotting the diffusion time, as a function of the detection volume, namely the diffusion-law, it is possible to identify the modality of mobility of the molecule[31,53,54].

### Stress granules isolation

About $2 \times 10^6$ SK-N-BE cells FUS^WT or FUS^P525L were plated on a 60 mm plate and transfected with 4 μg of vectors expressing sh1 and 4 μg of sh2 (TRCN0000289812, TRCN0000289814). The day after, cells were transferred to a 150-mm plate. When required, 96 h after transfection, FUS expression was induced with 50 ng/ml doxycycline for 24 h, stressed 1 h before harvesting, and harvested according to the protocol described by ref. 19. In order to normalize the immunoprecipitation efficiency, we used the invariant transcript CALM1 as an endogenous control in the comparison of different SG isolation experiments.

### SG RNA-seq analysis

Purified SG and relative inputs were generated in Dox- ($n = 4$ biologically independent replicates); upon FUS^P525L overexpression of ($n = 2$ biologically independent replicates) and FUS^WT ($n = 2$ biologically independent replicates) in condition of METTL3 depletion. RNA libraries for all samples were produced using Stranded Total RNA Prep with Ribo-Zero Plus (Illumina). All samples were sequenced on an Illumina Novaseq 6000 Sequencing system. Trimmomatic (v0.39)[56] and Cutadapt (v3.2)[57] were used to remove adapter sequences and poor-quality bases; the minimum read length after trimming was set to 35. Reads aligning to rRNAs were filtered out; this first alignment was performed using Bowtie2 software (v2.4.2) (https://bowtie-bio.sourceforge.net/bowtie2/index.shtml). STAR software (v2.7.7a)[58] was used to align reads to GRCh38 genome using ENCODE standard parameters referred in the manual. PCR duplicates were removed from all samples using MarkDuplicates command from Picard suite (v2.24.1) (https://broadinstitute.github.io/picard/). Uniquely mapping fragments were counted for each annotated gene (Ensembl release 99) using Htseq software (v0.13.5)[59]. EdgeR R package (v3.34.1)[60] was used to compare SG-enriched RNAs to their relative input samples. RNAs with $\log_2 FC > 1$ and FDR <0.05 were defined as "enriched"; those with $\log_2 FC < -1$ and FDR <0.05 were defined as "depleted"; all the other were labeled as "invariant". Differential enrichment analysis was performed using edgeR software defining the contrasts: (IP1-INP1)-(IP2-INP2) for the compared conditions. The significance threshold was set to FDR <0.05. Moreover, only RNAs expressed more than 1 FPKM in at least one sample type (IP1 or INP1 or IP2 or INP2) and that resulted as SG "enriched" in one of the analyzed conditions were defined as differentially enriched. Differential enrichment was taken into consideration in order to define the ΔMETTL3-enriched and ΔMETTL3-depleted groups. In all the analyses that use transcript sequences, fasta files were retrieved from Ensembl using biomart[61]. In case of genes with multiple isoforms a representative isoform was selected (the longest isoform with the same biotype of the gene was selected as representative; the longest isoform for non-coding RNAs). In order to overcome the high variability related to immunoprecipitation-based experiments, we performed an Enrichment convergence analysis. For each analyzed condition, we first ranked the expressed RNAs by their SG enrichment ($\log_2 FC$) and then we selected a fixed number of enriched RNAs for each dataset and compared these three equally sized RNA sets. We repeated this analysis selecting different fixed numbers of enriched RNAs, gradually reducing the inclusion of the noise from the background transcriptome (fixed numbers of top enriched RNAs: 5000, 4500, 4000, 3500, 3000, 2500, 2000, 1500, 1000, 500).

To avoid the technical bias due to the high variability of the library size of IP samples, we performed Resampling analysis in different conditions. We randomly sampled fragments from the alignment files of the other conditions in order to mirror the library size of the FUS^P525L condition experiment, used as reference, and we repeated enrichment analyses. Random sampling of fragments was performed with Picard suite using FilterSamReads function and using random subsets of fragments ids list as input (https://broadinstitute.github.io/picard).

### K-mers analysis

The average transcript frequency of each k-mer was calculated for SG-enriched RNAs and background distribution (*invariant* group), and their ratio (fold-change, FC) was computed. Then, the $\log_2 FC$ of k-mers with $k = 4$ of all the analyzed conditions were used for Heatmap representation applying k-means clustering ($n$ clusters = 3). Heatmaps graphical representations were depicted using the ComplexHeatmap R package (v2.8.0) (https://bioconductor.org/packages/release/bioc/html/ComplexHeatmap.html).

### meRIP-seq and m⁶A-CLIP

meRIP-seq was performed on FUS^P525L SK-N-BE cells as described by Dominissini et al.[62] with some adjustments. PolyA+ selection was performed thanks to the Poly(A)Purist-MAG kit (Invitrogen, #AM1922). PolyA+ RNA was incubated 4 min at 94 °C for fragmentation and 5 μg of fragmented polyA+ RNA was used for the immunoprecipitation with the anti-m⁶A polyclonal antibody (Abcam, #ab151230). The elution of the immunoprecipitated fraction was carried out thanks to the protocol described in ref. 63. M⁶A-CLIP for qRT-PCR validation of the meRIP-seq was performed as described in ref. 52.

### MeRIP-seq analysis

RNA libraries for all samples were produced using Stranded Total RNA Prep with Ribo-Zero Plus (Illumina). All samples were sequenced on an Illumina Novaseq 6000 Sequencing system with an average of about 74 million 100 nt long paired-end read pairs. Trimmomatic (v0.39)[56] and Cutadapt (v3.2)[57] were used to remove adapter sequences and poor-quality bases; the minimum read length after trimming was set to 35. Using Bowtie2 software (v2.4.2)[64] reads aligned to contaminant sequences of human rRNAs retrieved from NCBI were discarded. STAR software (v2.7.7a)[58] was used to align reads to the GRCh38 genome using ENCODE standard parameters referred in the manual. PCR duplicates were removed from all samples using MarkDuplicates command from Picard suite (v2.24.1) (https://broadinstitute.github.io/picard/) and multi-mapped and unproperly paired reads were filtered out using bamtools (v2.5.1)[65] and samtools (v1.7)[66] respectively. Peak calling was performed using exomePeak2 (v1.9.1) (https://github.com/ZW-xjtlu/exomePeak2) software filtering out peaks with $\log_2 FC$ (m⁶A-IP/input) <1; adjusted $p$ value >0.01 and with less than a total of ten fragments in input or m⁶A-IP samples. Peaks were annotated with Ensembl reference gtf (release 99) using bedtools (v2.29.1) with $-s$ $-split$ $-wao$ parameters. After peak annotation, isoform and gene expression was assessed using RSEM (v1.3.1)[67] on input samples. Peaks overlapping genes expressed <1 FPKM in the input condition were filtered out. Meta-gene peak coverage and motif enrichment analysis was performed with RNAmod[68].

Reads coverage for the forward and reverse strand were generated using deepTools bamCoverage software (v3.5.1; https://deeptools.readthedocs.io/en/develop/content/tools/bamCoverage.html).

### FUS HITS-CLIP

FUS HITS-CLIP was performed according to the protocol described by ref. 69 with some adjustments. In particular, $10^6$ SK-N-BE cells were

exposed to 50 ng/mL Doxycycline (Sigma-Aldrich, #D9891) for 24 h. Cells were UV-C crosslinked and lysed. The extract was treated with Rnase I (1:200, Life technologies, #AM2295) and Turbo Dnase (Life technologies #AM2238) for 3 min at 37 °C. About 2–3 mg of extract were diluted 2 mg/ml with lysis buffer and incubated with 100 µl of anti-FLAG beads (Sigma, # M8823) for 2 h at 4 °C. Beads were washed according to ref. 69 and resuspended in 190 µl of PK buffer added with 10 µl of Proteinase K (Thermo Fisher Scientific, #YSJ-762-Q) and incubated for 20 min shaking at 1100 at 37 °C. After this step, 200 µl of PK buffer + 7 M urea were added and incubated for 20 min at 1100 rpm at 37 °C. RNA was extracted with TRIzol reagent (Invitrogen, #15596026) according to the manufacturer's instructions for RNA-seq.

### HITS-CLIP analysis

RNA libraries were produced using Stranded Total RNA Prep with Ribo-Zero Plus (Illumina). All samples were sequenced on an Illumina Novaseq 6000 Sequencing system with an average of about 44,974,821 million 100 nt long paired-end read pairs. Trimmomatic (v0.39)[56] and Cutadapt (v3.2)[57] were used to remove adapter sequences and poor-quality bases; the minimum read length after trimming was set to 35. Using Bowtie2 software (v2.4.2)[64] reads aligned to contaminant sequences of human rRNAs, RN7SL and 7SK retrieved from NCBI or Ensembl were discarded. STAR software (v2.7.7a)[58] was used to align reads to the GRCh38 genome using the following parameters: --readFilesCommand zcat --outSAMtype BAM Unsorted --alignEndsType EndToEnd --outFilterMultimapNmax 10 --outFilterIntronMotifs RemoveNoncanonical --alignSJoverhangMin 12 --outFilterMatchNmin 15 --outFilterMismatchNoverLmax 0.05 --outFilterMultimapScoreRange 3 --alignIntronMax 20000 --seedMultimapNmax 200000 --seedPerReadNmax 30000 --peOverlapNbasesMin 35 --outSAMattributes NH HI NM MD AS nM jM jI XS. PCR duplicates were removed from all samples using MarkDuplicates command from Picard suite (v2.24.1) (https://broadinstitute.github.io/picard/) and multi-mapped and unproperly paired reads were filtered out using bamtools (v2.5.1)[65] and samtools (v1.7)[66], respectively. Peak calling was performed using standard settings of OmniCLIP software (v0.2.0)[70] and selecting only significant peaks (Bonferroni corrected $p$ value <0.05). Bigwig files of reads coverage for the forward and reverse strand were generated using bamCoverage function (v3.5.1) of deepTools suite (https://deeptools.readthedocs.io/en/develop/) considering only the first read of the pairs.

For HITS-CLIP and meRIP-Seq experiments, the visual inspection of signal in peak regions was accomplished using IGV tool (v2.8.13). For each replicate of both experiments bigwig normalization was performed using bigwigCompare function (v3.5.1) from deepTools suite (https://deeptools.readthedocs.io/en/develop/) subtracting input signal to the IP one while Heatmap representation of experiment signal was performed using computeMatrix and plotHeatmap functions.

### GO analysis

Gene ontology analysis was performed using WebGestalt R tool (v0.4.4)[71] applying weighted set cover reduction.

### Reporting summary

Further information on research design is available in the Nature Portfolio Reporting Summary linked to this article.

## Data availability

The data supporting the findings of this study are available from the corresponding authors upon request. The high throughput sequencing data generated in this study have been deposited in the GEO database under accession code GSE242771. Source data are provided with this paper.

## Code availability

All software, links to websites, or tools used for this work are referred to in the methods section or in the figure legends. Additional dedicated scripts developed for this work are available upon request.

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

## Acknowledgements

We thank M. Morlando, A. Fatica, V. De Turris, A. Rosa, J. Martone, J. Rea, and A. Colantoni for useful discussions and suggestions. We also thank F. Margarita and M. Marchioni for technical help and M. Caruso for assistance. This work is dedicated to the memory of Dr. Silvia Biscarini, whose legacy of dedication to science and teaching will forever remain a great example. This work was partially supported by grants from ERC-2019-SyG 855923-ASTRA, AIRC IG 2019 Id. 23053, and PRIN 2017 2017P352Z4 to I.B.; "National Center for Gene Therapy and Drug based on RNA Technology" (CN00000041) and NextGenerationEU PNRR MUR to I.B. and G.V.; ERC-2018-CoG 818669-BrightEyes to G.V.; "Sapienza" Ateneo Project 2021 n. RM12117A61C811CE and Regione Lazio PROGETTI DI GRUPPI DI RICERCA 2020 - A0375-2020-36597, NextGenerationEU through the Italian Ministry of University and Research under PNRR - M4C2-I1.3 Project PE_00000019 "HEAL ITALIA" CUP (B53C22004000006) to D.R.; Sapienza departmental projects 2023 (RD12318A998C70B8) to G.D.T.

## Author contributions

G.D.T., A. Giuliani, and I.B. designed and conceived the study. The experiments were performed and analyzed by G.D.T., A. Giuliani., M.C.B., M.L., T.S., A. Grandioso, D.M., F.C., E.P., S.Z., G.V., D.R., M.S., and S.L. Bioinformatics data analysis was performed by A.S. The original draft of the manuscript was written by I.B., G.D.T., and A. Giuliani with suggestions from all the other authors. I.B. supervised the project.

## Competing interests

Sapienza University of Rome, Fondazione Istituto Italiano di Tecnologia (IIT) are currently in the process of a patent application. Patent applicant: Sapienza University of Rome, Fondazione Istituto Italiano di Tecnologia (IIT) name of inventors: Irene Bozzoni, Gaia Di Timoteo, Andrea Giuliani, Adriano Setti application number: IT102023000022302 status of application: pending specific aspect of manuscript covered in patent application: use of METTL3 inhibitors in the treatment of protein aggregation diseases. The remaining authors declared no competing interests.
