## [Peer Review File · Nature Communications]

M6A reduction relieves FUS-associated ALS granulesREVIEWER COMMENTS

Reviewer #1 (Remarks to the Author):

This manuscript addresses the mechanism by which pathogenic FUS mutations alter the composition of stress granules. The work largely builds on a bioarchive preprint that documents a difference in stress granule composition between cells with WT FUS and the P525L pathogenic mutation. The demonstration that m6A modification changes the composition in a manner dependent on the pathogenic FUS mutation is interesting and when sufficiently developed could be published in Nature Communications. However, before the manuscript is considered further for publication the issues below would need to be addressed.

This review is from Roy Parker and I would be willing to clarify these comments for the authors directly if needed.

Specific Issues:

1) The starting point for this manuscript is that there is a difference in the transcriptome of stress granules with WT FUS or the FUS P525L mutant. This work is then followed up by demonstrating a difference in the stress granule transcriptomes with the FUS P525L mutation, when METTL3 is inhibited. I recommend the authors strengthen these conclusions (in both the preprint, and this manuscript) in two manners.

a) First, one needs to determine if the differences in the mRNAs in the stress granules are simply caused by differences in expression or are actually differentially localized. The authors can make this clear by only examining mRNAs that are expressed over a certain level in all the relevant cell lines (perhaps they have already done so, but this is not explicitly stated). This should be clarified.

b) Second, given the complexity of the stress granule purification methods, I would consider the conclusion much stronger if the authors validated the differences between WT and mutant FUS, or +/- METTL3 function by smFISH of some of the key mRNAs they predict to be different. This is a straightforward experiment that would demonstrate the same result by an orthogonal method, thereby greatly strengthening the conclusion.

2) In figure 1, I recommend the authors also show the comparison of WT and mutant FUS (in the same style as 1b,c,d). I realize this is the point of the other manuscript, but such a demonstration here is also critical to this manuscript.

3) The authors might like to be aware of, and discuss, the work of Yoneda et al (PMID: 34681673), who argued FUS preferentially binds m6A mRNAs, and that the retention of mutant FUS in cytosolic aggregates could be relieved by transfection with short m6A containing RNAs.

4) As it stands now, the manuscript describes an interesting observation but does not address the molecular basis for that difference. The work would be more similar to manuscripts published in Nature Communications if additional insight into the mechanism by which mutant FUS affects the composition of stress granules was provided.

5) Although I would not require it for publication, the demonstration that chemical inhibition of m6A methylation could rescue FUS toxicity in an animal model of disease would make this work a major contribution.

Reviewer #2 (Remarks to the Author):

In this manuscript, the authors claimed that the decrease in m6A modification by the knockdown of METTL3 restored phenotypes observed in the ALS cellular models containing the mutant FUS. These phenotypes include the RNA composition of stress granules (SGs) as well as their numbers and recovery rates. They first purified stress granules in the ALS cellular models through immunoprecipitation and analyzed the RNA compositions of SGs. The ALS cellular models with mutant FUS showed highly distinct RNA content when compared to the control cells with WT FUS. When the METTL3 level was reduced with shRNAs, they found that the RNA composition in the mutant FUS cells became closer to the one with WT FUS. Similarly, the number of SGs, increased in the ALS models, was reduced upon the METTL3 knockdown. The rate of SG dissolution, upon stress removal, was also restored when the METTL3 level was reduced. Similar phenotypic reliefs were observed when METTL3 was chemically inhibited. Although presented data are interesting, the paper needs to be improved in multiple aspects before consideration for publication in Nature Communications:

1. To analyze the RNA content of SGs, the authors purified SGs using the antibody against GFP-tagged G3BP1. Notably, SGs have a core-shell architecture and the SG core persists after cell lysis while the shell part dissolves (PMID: 26777405). In addition, protein droplets of the purified FUS ALS mutants tend to have more solid-like characteristics, compared to WT FUS. These points raise a possibility where the measured RNAs in this manuscript may reflect the content of the core part, rather than the full transcriptome of the SG. Thus, it will be very important to verify their results with RNA FISH experiments. I strongly recommend that from four different groups of transcripts ("commonly enriched", "delMETTL3-enriched", etc) as well as those in "relocated RNAs", several RNAs should be randomly selected and tested for their localization with respect to SGs.
2. METTL3 downregulation restored the RNA composition of SGs in ALS models. However, the authors did not show changes of m6A levels in the individual RNAs upon the knockdown of METTL3. In Fig. 1e, the percentages of m6A-containing RNAs are shown for each group. But, do you see a strong reduction of m6A modification for those RNAs relocated, compared to other groups, in the METTL3 knockdown? Since the total m6A level in the entire RNA was reduced only by ~ 20% (Fig. S1e), it is likely that the extent of m6A changes in individual RNAs may not be large. In other words, the authors should provide extra evidence for the origin of changes in SG transcriptome; is it directly related to the altered m6a level of the given RNA or is it an indirect effect?
3. The manuscript remains highly phenomenological, not providing mechanistic insights. How is the altered RNA content related to stress granule number/recovery? What is the model connecting the decrease in m6a level with these phenotypes? Although presented data are interesting, mechanistic understanding is currently lacking.
4. Figure quality should be improved; fonts are way too small; sometimes too many subfigures.
5. For FCS data, further technical details should be specified. The full autocorrelation curves should be provided for each construct/condition. In Methods, it is stated that FCS measurements were conducted in the cytoplasm or inside SGs. Were parameters such as diffusion coefficients and confinement strengths similar regardless of the subcellular locations? It will be helpful to include more explanation on the confinement strength in the FCS analysis. Is it possible to distinguish changes in oligomerization status versus the degree of confinement using FCS data? Comparing data in Fig 4 and S4, the diffusion coefficient of G3BP1 is about 10-fold higher than that of FUS in recovery. What the origin of this large difference?
6. In Fig. S1o, it doesn't seem that the results from RT-qPCR were directly compared to the MeRIP-seq?
7. Comparing numbers in Fig. B-D and Fig. S1H-I, they don't seem to match with one another. For

example, there are total 1637 (1176 + 461) transcripts enriched in SG for FUS(P525L) upon the knockdown of METTL3. But, in Fig. S1i, the same corresponding category contains 2874 transcripts. The discrepancy might be a natural outcome of analysis, but causes confusion.

8. For those RNAs in Fig. S2h, provide how m6A levels are changed for each RNA under different experimental conditions. This will help convince that the origin of relocation is indeed due to the change in the m6A level of individual RNAs.

9. The authors stated that in FUS(P525L) cells, granules with FUS-only signals were observed after recovery. However, this is not clearly visible in Fig. S3a. It will be good to have granules with FUS only signals highlighted with arrows.

Reviewer #3 (Remarks to the Author):

Timoteo et al., presents work showing how m6A-tagging of RNAs influences the localization of RNAs into ALS mutant FUS granules. Genetic depletion of METTL3 or through a METTL3 inhibitor STM-2457 reveal that mutant FUS granule RNA composition reverts back to WT FUS granule which they show by GFP-G3BP enrichment followed by sequencing. Furthermore, Timoteo et al. show that mutant FUS, but not WT FUS granule formation is affected by m6A levels. Indeed, they report that the number of condensates and phase separation properties of mutant FUS granules differs from that of WT FUS granules. Much like RNA composition, depletion of m6A levels specifically triggers the reversion mutant FUS granule biophysical properties to WT FUS. They extended this analysis to various neuronal and fibroblast c from different ALS-related FUS patient cell lines to demonstrate that the relationship between m6A and mut FUS is a general phenomenon. Interestingly, depletion of m6A levels did not alter WT FUS granule behavior, which is consistent with previous reports indicating that changes in m6A levels had no effect in stress granule formation. Taken together, Timoteo et., show an interesting relationship between levels of m6A and FUS granules, which is specific to ALS-related mutations and speculate that STM-2457 may be a therapeutic avenue to alleviate ALS pathogenicity caused by FUS mutations. The data/observations are interesting and potentially significant for the field of ALS and RNA regulation. Although the authors state that more work needs to be done in their discussion, the mechanisms underlying how m6A regulates mut FUS granule formation is not explored in this work, decreasing the impact of the manuscript.

Comments:

The SG seq experiment is done with an N=2 replicate per experimental condition (WT vs FUS P525L) which may be an issue for the robustness of this experiment.

Is there functional enrichment for the RNAs within mut FUS SG that are affected by METTL3 KD and are RNAs somehow linked to the known pathomechanisms of ALS? What about the m6a-ylated RNAs in the FUS P525L identified by meRIP-seq?

The authors performed meRIP-seq on mut FUS but not on WT FUS cells. It would have been informative to know whether expression of mut FUS have an effect on m6a-tagging, compared to WT FUS.

The authors made many interesting observations on the granule dynamics of mut FUS and what happens to it when global m6A levels are altered, however, how ALS-mutant FUS granules are influenced by m6A are not addressed by the authors.

Is FUS able to bind m6a tagged RNAs? The Barmada group has reported that TDP-43 is able to bind m6a tagged RNAs, is this also true for FUS? If so, how does ALS mutations affect this process

If global decrease of m6A levels affect mut FUS granule morphology, does global increase of m6a via deletion of ALKBH5, for instance, have the opposite effect on mut FUS granules morphology and dynamics?

The proposal that decreasing m6A levels will be beneficial for ALS is somewhat contrary to the findings that METTL3/14 is important for boosting axonal regeneration (10.1016/j.neuron.2017.12.036). Perhaps in ALS m6A is hyperactivated and dampening it is beneficial which is indicated in the recent work of the Barmada Group (10.1016/j.molcel.2022.12.019). These should at least be discussed by the authors.

Other comments:

There are many figures that are very, very blurry.

Reviewer #1

This manuscript addresses the mechanism by which pathogenic FUS mutations alter the composition of stress granules. The work largely builds on a bioarchive preprint that documents a difference in stress granule composition between cells with WT FUS and the P525L pathogenic mutation. The demonstration that m6A modification changes the composition in a manner dependent on the pathogenic FUS mutation is interesting and when sufficiently developed could be published in Nature Communications. However, before the manuscript is considered further for publication the issues below would need to be addressed.

This review is from Roy Parker and I would be willing to clarify these comments for the authors directly if needed.

Specific Issues:

1) The starting point for this manuscript is that there is a difference in the transcriptome of stress granules with WT FUS or the FUS P525L mutant. This work is then followed up by demonstrating a difference in the stress granule transcriptomes with the FUS P525L mutation, when METTL3 is inhibited. I recommend the authors strengthen these conclusions (in both the preprint, and this manuscript) in two manners.

a) First, one needs to determine if the differences in the mRNAs in the stress granules are simply caused by differences in expression or are actually differentially localized. The authors can make this clear by only examining mRNAs that are expressed over a certain level in all the relevant cell lines (perhaps they have already done so, but this is not explicitly stated). This should be clarified.

We thank the reviewer for allowing us to clarify this important point. As now indicated also in Mariani's paper (doi: <https://doi.org/10.1101/2023.09.11.557245>) the comparison of the WT and mutant conditions show no differences in the input samples demonstrating that the differential RNA enrichment observed in SGs is not due to differential gene expression. We have now included these data in the text (New Fig. S1m). Furthermore, following the reviewer suggestion of examining mRNAs that are expressed over a certain level, we stratified the SK-N-BE transcriptome in 5 groups based on their average FPKM (low, medium-low, medium, medium-high and high). As displayed in the new Fig. S1n, both the commonly enriched and the Δ METTL3-enriched RNAs are composed by 70% of species belonging to the medium expression group (from 2 to 18 FPKM), reinforcing the data that the expression level does not affect our observations.

b) Second, given the complexity of the stress granule purification methods, I would consider the conclusion much stronger if the authors validated the differences between WT and mutant FUS, or +/- METTL3 function by smFISH of some of the key mRNAs they predict to be different. This is a straightforward experiment that would demonstrate the same result by an orthogonal method, thereby greatly strengthening the conclusion.

We agree with the reviewer that this is an important control. In line with this request, we selected a few mRNAs predicted to be differently enriched across the different conditions considered in our study and validated their localization by smFISH analysis (new Fig.2b and Fig. S2i). We selected CLSTN1 as a representative of the relocated RNAs group, GAPDH of the depleted species, CALM1 of the invariant and HUWE1 of the commonly enriched. As indicated in Fig. 2b and Fig. S2i, the CLSTN1 mRNA shows a clear re-localization in SG upon METTL3 downregulation, while the other controls behave as expected, with GAPDH being always excluded from SG, CALM1 as unchanged in all conditions and HUWE1 always enriched in SG.

2) In figure 1, I recommend the authors also show the comparison of WT and mutant FUS (in the same style as 1b,c,d). I realize this is the point of the other manuscript, but such a demonstration here is also critical to this manuscript.

We have added the comparison of WT and mutant FUS in figure S1, as recommended (Fig. S1h).

3) The authors might like to be aware of, and discuss, the work of Yoneda et al (PMID: 34681673), who argued FUS preferentially binds m6A mRNAs, and that the retention of mutant FUS in cytosolic aggregates could be relieved by transfection with short m6A containing RNAs.

We appreciate the input provided by the reviewer since it has allowed us to better define the relationship between FUS binding and m⁶A.

Yoneda et al. demonstrated that the presence of m⁶A on a specific 13nt long RNA enhances FUS binding, and that the transfection of such fragments relieves FUS aggregates in hyperosmotic stress conditions. We think it is difficult to compare Yoneda's data with ours for several reasons: the first is the general consideration that a clear definition of whether m⁶A increases or inhibits FUS interaction to RNA has not yet been clarified. Indeed, the same group previously showed that m⁶A can also interfere with FUS binding (J Biol Chem. 2020, 295:5626. "Knockdown of METTL3 or YTHDC1 also enhanced the interaction of pncRNA-D with TLS..."). The second reason relates to the short size of the RNA used in their study and to the fact that it was specifically selected in vitro for binding to FUS. Endogenous RNAs are much more complex in terms of interactions with many different proteins, in addition to FUS, that could control RNA exclusion or recruitment to SGs, and m⁶A certainly adds another layer of regulation whose effects are likely to be context dependent.

Finally, considering the sequence used by Yoneda, it is very much plausible that such a kind of FUS "aptamer" could likely interfere with granule formation.

However, given the interest in this important comment, and in order to provide some further clarification, we performed HITS-CLIP for FUS and compared the fraction of RNA interactors excluded or included in SG (see figure below).

We found that among the direct FUS interactors, only a small fraction (8%) was methylated (new Fig. S1w). Moreover, peaks analysis showed that the majority of FUS binding sites do not overlap with m⁶A sites (new Fig. S1x) and resulted depleted for DRACH motif (new Fig. S1y), indicating that m⁶A target regions per se are not preferred sites for FUS binding. These data are in agreement with the observation that methylated species are preferentially excluded from FUS^{P525L} SG ("loss" species) and only reimported upon METTL3 downregulation ("relocated" species). Overall, these data allowed us to conclude that in ALS conditions m⁶A-enriched species are less recruited into the SG and are not directly bound by FUS.

4) As it stands now, the manuscript describes an interesting observation but does not address the molecular basis for that difference. The work would be more similar to manuscripts published in Nature Communications if additional insight into the mechanism by which mutant FUS affects the composition of stress granules was provided.

Solicited by this important comment of the reviewer, we performed some additional experiments in order to provide more insights on the molecular basis of the observed phenomenon.

In particular:

- 1) We carried out the analysis of the m⁶A levels in control and mutant cells. We found that FUS^{P525L} cells are characterized by high levels of m⁶A and that METTL3 downregulation re-established them similar to control levels (new Fig. S2u). Such finding well explains the results obtained, according to which the reduction of m⁶A levels in FUS^{P525L} cells restores conditions similar to control both in terms of the RNA content of SG as well as their dynamics. These data underline the importance of m⁶A homeostasis, together with ALS-associated mutations, in triggering phase transition of pathological aggregates.*
- 2) To support the hypothesis that hypermethylation is responsible for the mis-regulated granules dynamics in ALS cells, we also increased m⁶A levels in our cellular systems either overexpressing the writer METTL3 or downregulating the eraser ALKBH5. As shown in new fig. 2e and 3c, both approaches increased the number of granules and decreased the recovery capacity in FUS^{P525L} cells. Notably, the same effects were reproduced also in FUS^{WT} cells.*
- 3) As indicated above, the new HITS-CLIP data show that the majority of FUS binding sites do not overlap with m⁶A sites and resulted depleted for DRACH motif, indicating that m⁶A target regions are not preferred sites for FUS binding. This is in agreement with the data showing that in ALS conditions, where m⁶A levels are higher, the m⁶A-enriched species are excluded from SG and are not bound by FUS.*

In conclusion, we provided additional insights demonstrating that m⁶A levels are increased in ALS, and that this increase can be linked to the presence of FUS. Furthermore, we showed that RNA hypermethylation can worsen the altered SG dynamics increasing granules number and decreasing SG recovery even in control cells.

5) Although I would not require it for publication, the demonstration that chemical inhibition of m⁶A methylation could rescue FUS toxicity in an animal model of disease would make this work a major contribution.

We agree with the reviewer that this would be a very relevant experiment. We are currently searching for such models; however, it is something not feasible in a short time. We will certainly plan to do it in the near future.

Reviewer #2 (Remarks to the Author):

In this manuscript, the authors claimed that the decrease in m⁶A modification by the knockdown of METTL3 restored phenotypes observed in the ALS cellular models containing the mutant FUS. These phenotypes include the RNA composition of stress granules (SGs) as well as their numbers and recovery rates. They first purified stress granules in the ALS cellular models through immunoprecipitation and analyzed the RNA compositions of SGs. The ALS cellular models with mutant FUS showed highly distinct RNA content when compared to the control cells with WT FUS. When the METTL3 level was reduced with shRNAs, they found that the RNA composition in the mutant FUS cells became closer to the one with WT FUS. Similarly, the number of SGs, increased in the ALS models, was reduced upon the METTL3 knockdown. The rate of SG dissolution, upon stress removal, was also restored when the METTL3 level was reduced. Similar phenotypic reliefs were observed when METTL3 was chemically inhibited. Although presented data are interesting, the paper needs to be improved in multiple aspects before consideration for

publication in Nature Communications:

1. To analyze the RNA content of SGs, the authors purified SGs using the antibody against GFP-tagged G3BP1. Notably, SGs have a core-shell architecture and the SG core persists after cell lysis while the shell part dissolves (PMID: 26777405). In addition, protein droplets of the purified FUS ALS mutants tend to have more solid-like characteristics, compared to WT FUS. These points raise a possibility where the measured RNAs in this manuscript may reflect the content of the core part, rather than the full transcriptome of the SG. Thus, it will be very important to verify their results with RNA FISH experiments. I strongly recommend that from four different groups of transcripts (“commonly enriched”, “delMETTL3-enriched”, etc) as well as those in “relocated RNAs”, several RNAs should be randomly selected and tested for their localization with respect to SGs.

Thanks to the reviewer for this comment. As indicated for ref#1, we selected a few mRNAs predicted to be differently enriched across the different conditions considered in our study and validated their localization by smFISH analysis (new Fig.2b and Fig. S2i). We used CLSTN1 as a representative of the relocated RNAs group, GAPDH of the depleted species, CALM1 of the invariant and HUWE1 of the commonly enriched. As indicated in Fig. 2b and Fig. S2i, the CLSTN1 mRNA shows a clear relocalization in SG upon METTL3 downregulation, while the other controls behave as expected, with GAPDH being always excluded from SG, CALM1 as unchanged in all conditions and HUWE1 always enriched in SG.

2. METTL3 downregulation restored the RNA composition of SGs in ALS models. However, the authors did not show changes of m6A levels in the individual RNAs upon the knockdown of METTL3. In Fig. 1e, the percentages of m6A-containing RNAs are shown for each group. But, do you see a strong reduction of m6A modification for those RNAs relocated, compared to other groups, in the METTL3 knockdown? Since the total m6A level in the entire RNA was reduced only by ~ 20% (Fig. S1e), it is likely that the extent of m6A changes in individual RNAs may not be large. In other words, the authors should provide extra evidence for the origin of changes in SG transcriptome; is it directly related to the altered m6a level of the given RNA or is it an indirect effect?

As requested by the reviewer, we have analyzed the extent of m⁶A changes in individual RNAs by m⁶A CLIP assay. We looked at the relocated class since it exhibited the highest difference in SG association between FUS^{WT} and FUS^{P525L} cells. As shown in new Fig. S2j, in the m⁶A-containing fraction we found a decrease of relocated RNAs upon METTL3 knock-down comparable to that of the WTAP mRNA which does not change localization upon m⁶A depletion. These data indicate that the effect of METTL3 knock-down is global and that, like all the other species, also relocated RNAs are hypo-methylated in these conditions.

3. The manuscript remains highly phenomenological, not providing mechanistic insights. How is the altered RNA content related to stress granule number/recovery? What is the model connecting the decrease in m6a level with these phenotypes? Although presented data are interesting, mechanistic understanding is currently lacking.

These are important comments. As also done in response to reviewer #1 we have added several experiments aimed at providing more mechanistic insides on the molecular basis of the observed phenomenon. In particular:

- 1) *We carried out the analysis of the m⁶A levels in control and mutant cells. We found that FUS^{P525L} cells are characterized by high levels of m⁶A and that METTL3 downregulation re-established them similar to control levels (new Fig. S2u). Such finding well explains the results obtained, according to which the reduction of m⁶A levels in FUS^{P525L} cells restores conditions similar to control both in terms of the RNA content of SG as well as their dynamics. These data underline the importance of m⁶A homeostasis, together with ALS-associated mutations, in triggering phase transition of pathological aggregates.*
- 2) *To support the hypothesis that hypermethylation is responsible for the mis-regulated granules dynamics in ALS cells, we also increased m⁶A levels in our cellular systems either overexpressing the writer METTL3 or downregulating the eraser ALKBH5. As shown in new fig. 2e and 3c, both approaches increased the number of granules and decreased the recovery capacity in FUS^{P525L} cells. Notably, the same effects were reproduced also in FUS^{WT} cells.*
- 3) *As indicated above, the new HITS-CLIP data show that the majority of FUS binding sites do not overlap with m⁶A sites and resulted depleted for DRACH motif, indicating that m⁶A target regions are not preferred sites for FUS binding. This is in agreement with the data showing that in ALS conditions, where m⁶A levels are higher, the m⁶A-enriched species are excluded from SG and are not bound by FUS.*

In conclusion, we provided additional insights demonstrating that m⁶A levels are increased in ALS, and that this increase can be linked to the presence of FUS. Furthermore, we showed that RNA hypermethylation can worsen the altered SG dynamics increasing granules number and decreasing SG recovery even in control cells.

4. Figure quality should be improved; fonts are way too small; sometimes too many subfigures.

We thank the reviewer for the attention paid also to this aspect. Even though the figures were designed and reported according to the journal guidelines, we tried to improve them. Unfortunately, also in consideration of the conspicuous number of new experiments, we were unable to reduce the total number of panels.

5. For FCS data, further technical details should be specified. The full autocorrelation curves should be provided for each construct/condition.

We agree with the reviewer about having a representative autocorrelation curve for each measured condition. We added the supplementary figure S4I with the autocorrelation curves for the central pixel, the sum of the inner 3x3 detector pixels and the sum of all detector pixels, for each cellular condition before the stress, upon the stress and during the recovery.

In Methods, it is stated that FCS measurements were conducted in the cytoplasm or inside SGs. Were parameters such as diffusion coefficients and confinement strengths similar regardless of the subcellular locations?

Yes, in principle FCS measurements will be different if different subcellular location are probed. However, for this study we always compared FCS measurements within SGs; only in recovery conditions we performed measures also outside the granules since G3BP1 is released free from them. Indeed, in this case we found higher D values. We think that such increase can be attributed to the fact that we conducted the measurements in the cytoplasm where the protein is in a free state. We added this information in the material and methods section: "... Multiple planar positions in cells

were selected to probe different points. Measurements were performed within SGs in stress and in the cytoplasm in recovery conditions. The fluorescence intensity was acquired....”

It will be helpful to

include more explanation on the confinement strength in the FCS analysis. Is it possible to distinguish changes in oligomerization status versus the degree of confinement using FCS data?

As requested, we added details and references on the confinement strength parameter.

In principle, with conventional FCS is possible to distinguish changes in oligomerization status, but due to the high variability of biological processes typically it is avoided as it is a quantification heavily prone to artefacts. In fact, a change of a factor 2 in the MW (monomer to dimer) of the molecule is reflected by a change of a factor 1.26 on the diffusion coefficient which is typically lower than the variability measurement in one single condition (due to the high variability of the processes within single-cells and within different cells). According to this technical issue, we avoided the stoichiometry analysis.

Comparing data in Fig 4 and S4, the diffusion coefficient of G3BP1 is about 10-fold higher than that of FUS in recovery. What the origin of this large difference?

As indicated above, upon recovery G3BP1 is released free from granules at difference with FUS, which instead persists in aggregates. Therefore, the increase of the D value is expected for G3BP1 but not for FUS, which retain the same D value as in stress conditions. This specification is now included in the text.

6. In Fig. S1o, it doesn't seem that the results from RT-qPCR were directly compared to the MeRIP-seq?

We thank the reviewer for this comment. In order to compare meRIP-Seq and qRT-PCR we performed a Person's correlation of the RNA-Seq signal (fold IP/INP) and the % INP derived from qRT-PCR analysis. We obtained a high and significant correlation coefficient ($R=0.82$, $p\text{-value} = 0.024$). Additionally, we have added IGV screens that allow the visualization of the meRIP-Seq signal corresponding to the described peaks (new Fig. S1s).

7. Comparing numbers in Fig. B-D and Fig. S1H-I, they don't seem to match with one another. For example, there are total 1637 (1176 + 461) transcripts enriched in SG for FUS(P525L) upon the knockdown of METTL3. But, in Fig. S1i, the same corresponding category contains 2874 transcripts. The discrepancy might be a natural outcome of analysis, but causes confusion.

We confirm that the observed discrepancy is a natural outcome of the analysis. It arises from the fact that Δ METTL3-enriched or depleted RNAs are subsets of the SG-enriched groups and include only the RNAs that exhibit a statistically significant differential enrichment between the compared conditions. To prevent misunderstanding, we have included the total number of SG-enriched RNAs for each specific condition in scatterplots in Fig. 1b, c, d and Supplementary Figure S1h and S1g. The complete data description is provided in Supplementary Table 1.

8. For those RNAs in Fig. S2h, provide how m6A levels are changed for each RNA under different experimental conditions. This will help convince that the origin of relocation is indeed due to the change in the m6A level of individual RNAs.

We performed the requested experiments and provided the extra information as described in point 2 and inserted new Fig. S2j.

9. The authors stated that in FUS(P525L) cells, granules with FUS-only signals were observed after recovery. However, this is not clearly visible in Fig. S3a. It will be good to have granules with FUS only signals highlighted with arrows.

In according to the reviewer suggestion, we highlighted the granules with FUS-only signals with arrows (see Fig. S3a).

Reviewer #3 (Remarks to the Author):

Timoteo et al., presents work showing how m6A-tagging of RNAs influences the localization of RNAs into ALS mutant FUS granules. Genetic depletion of METTL3 or through a METTL3 inhibitor STM-2457 reveal that mutant FUS granule RNA composition reverts back to WT FUS granule which they show by GFP-G3BP enrichment followed by sequencing. Furthermore, Timoteo et al. show that mutant FUS, but not WT FUS granule formation is affected by m6A levels. Indeed, they report that the number of condensates and phase separation properties of mutant FUS granules differs from that of WT FUS granules. Much like RNA composition, depletion of m6A levels specifically triggers the reversion mutant FUS granule biophysical properties to WT FUS. They extended this analysis to various neuronal and fibroblast c from different ALS-related FUS patient cell lines to demonstrate that the relationship between m6A and mut FUS is a general phenomenon. Interestingly, depletion of m6A levels did not alter WT FUS granule behavior, which is consistent with previous reports indicating that changes in m6A levels had no effect in stress granule formation. Taken together, Timoteo et., show an interesting relationship between levels of m6A and FUS granules, which is specific to ALS-related mutations and speculate that STM-2457 may be a therapeutic avenue to alleviate ALS pathogenicity caused by FUS mutations. The data/observations are interesting and potentially significant for the field of ALS and RNA regulation. Although the authors state that more work needs to be done in their discussion, the mechanisms underlying how m6A regulates mut FUS granule formation is not explored in this work, decreasing the impact of the manuscript.

Comments:

The SG seq experiment is done with an N=2 replicate per experimental condition (WT vs FUS P525L) which may be an issue for the robustness of this experiment.

The RNA-seq data were validated with qRT-PCR for representative species of the different classes (Common, Loss, Relocated and Depleted); moreover, to corroborate the findings we also performed smFISH analysis on selected mRNA species which allowed us to validate their localization as predicted by the RNA-seq (see new Fig. 2b and Fig. S2i).

Is there functional enrichment for the RNAs within mut FUS SG that are affected by METTL3 KD and are RNAs somehow linked to the known pathomechanisms of ALS? What about the m6Aylated RNAs in the FUS P525L identified by meRIP-seq?

We have added the GO analysis of relocated RNAs within mut FUS SG. As shown in new Fig. S1u, this class includes several genes related to neuronal cellular components. Moreover, through literature survey we found that 200 over 461 genes have at least one previous publication linking them to Neurodegeneration or ALS (Supplementary Table 1). Interestingly, among these RNAs we found important genes such as CLSTN1, ULK1, NGFR and KIF5A. Coherently with the global percentage of methylated RNAs among the relocated transcripts (37%), also in the GO categories we

found similar fractions of m⁶A-containing RNAs (32%-40%). We provided the text with this information.

The authors performed meRIP-seq on mut FUS but not on WT FUS cells. It would have been informative to know whether expression of mut FUS have an effect on m⁶A-tagging, compared to WT FUS. The authors made many interesting observations on the granule dynamics of mut FUS and what happens to it when global m⁶A levels are altered, however, how ALS-mutant FUS granules are influenced by m⁶A are not addressed by the authors.

Is FUS able to bind m⁶A tagged RNAs? The Barmada group has reported that TDP-43 is able to bind m⁶A tagged RNAs, is this also true for FUS? If so, how does ALS mutations affect this Process. If global decrease of m⁶A levels affect mut FUS granule morphology, does global increase of m⁶A via deletion of ALKBH5, for instance, have the opposite effect on mut FUS granules morphology and dynamics? The proposal that decreasing m⁶A levels will be beneficial for ALS is somewhat contrary to the findings that METTL3/14 is important for boosting axonal (10.1016/j.neuron.2017.12.036). Perhaps in ALS m⁶A is hyperactivated and dampening it is beneficial which is indicated in the recent work of the Barmada Group (10.1016/j.molcel.2022.12.019). These should at least be discussed by the authors.

We thank the reviewer for this important comment that induced us to perform more experiments along this line. In summary, as also detailed for ref#1 and #2, we performed the following experiments:

1) We carried out the analysis of the m⁶A levels in control and mutant cells. We found that FUS^{P525L} cells are characterized by high levels of m⁶A and that METTL3 downregulation re-established them similar to control levels (new Fig. S2u). Such finding well explains the results obtained, according to which the reduction of m⁶A levels in FUS^{P525L} cells restores conditions similar to control both in terms of the RNA content of SG as well as their dynamics. These data underline the importance of m⁶A homeostasis, together with ALS-associated mutations, in triggering phase transition of pathological aggregates.

As pointed out by the reviewer, m⁶A is also important for boosting axonal growth. Since increased axonal growth was reported in multiple ALS models (10.1038/s42003-021-02538-8;10.1016/j.isci.2018.12.026; doi.org/10.1093/hmg/ddp534), m⁶A decrease could be beneficial also for this aspect.

2) To support the hypothesis that hypermethylation is responsible for the mis-regulated granules dynamics in ALS cells, we also increased m⁶A levels in our cellular systems either overexpressing the writer METTL3 or downregulating the eraser ALKBH5. As shown in new fig. 2e and 3c, both approaches increased the number of granules and decreased the recovery capacity in FUS^{P525L} cells. Notably, the same effects were reproduced also in FUS^{WT} cells.

3) As indicated above, the new HITS-CLIP data show that the majority of FUS binding sites do not overlap with m⁶A sites and are depleted for DRACH motif, indicating that m⁶A target regions are not preferred sites for FUS binding. This is in agreement with the data showing that in ALS conditions, where m⁶A levels are higher, the m⁶A-enriched species are excluded from SG and are not bound by FUS.

In conclusion, we provided additional insights demonstrating that m⁶A levels are increased in ALS, and that this increase can be linked to the presence of FUS. Furthermore, we showed that RNA hypermethylation can worsen the altered SG dynamics increasing granules number and decreasing SG recovery even in control cells.

Other comments:

There are many figures that are very, very blurry.

We thank the reviewer for the attention paid also to this aspect.

We tried to improve the figures.

** See Nature Portfolio's author and referees' website at www.nature.com/authors for information about policies, services and author benefits.

REVIEWER COMMENTS

Reviewer #1 (Remarks to the Author):

The authors have adequately addressed my earlier comments.

One item the authors might wish to consider is whether the CLSTN1 mRNA is enriching in P-bodies rather than stress granules. I suggest this possibility since the mRNA puncta in the smFISH tend to dock to the edges of stress granules, which is often where P-bodies would reside.

Reviewer #2 (Remarks to the Author):

The authors addressed several comments previously raised, but I still have several remaining concerns as elaborated below. Importantly, it is still unclear whether and how altered transcriptomes are linked to SG assembly and disassembly. The manuscript describes these two phenotypic behaviors upon the knockdown or inhibition of the m6A writer, but how are they related? The paper still feels phenomenological, and probing this connection would significantly increase its impact.

- The authors concluded that FUS P525L SG tends to exclude m6A-containing transcripts. If the fractions of m6A-containing transcripts are compared between SG-enriched vs depleted population in FUS P525L cells, I suspect that the m6A-containing fraction would still be higher for SG-enriched one? This might be because the length of transcript, which also tends to be correlated with the number of m6A sites, is a strong predictor of SG partitioning. But this analysis would potentially put a question mark on the author's conclusion. Also, what is the m6A characteristics of the transcripts in the "depleted" group, those originally enriched in FUS P525L SG but depleted after METTL3 knockdown? Do they exhibit an opposite trend from the "relocated" group, like a low fraction of m6A-containing transcripts?

- The overexpression of METTL3 increased the number of SGs even in FUS WT cells. Does it involve a change in the localization of FUS to the cytoplasm? Do these SGs show other phenotypes, such as FUS-only aggregates in recovery and reduced internal dynamics, similar to those in FUS P525L cells? The number of SGs can be affected by the expression levels of SG components (for example, overexpression of G3BP1 causes SG assembly in the absence of stress conditions; 10.1083/jcb.200502088). It is also possible that the dissolution takes longer simply because of the higher number of SGs assembled initially. Monitoring the full dissolution kinetics and the presence of irreversible species would be a better way of probing the recovery dynamics of SGs.

- Transcripts were classified into four groups based on differential enrichment in SG. My understanding is that "delMETTL3-depleted" group corresponds to transcripts originally enriched in SG but then depleted upon METTL3 knockdown. But, in smFISH data (line 199-202), GAPDH was classified into "depleted" and CALM1 into "invariant". Based on the definition and RNA localization data, shouldn't both of them correspond to "invariant"?

- For smFISH data, adding several zoomed-in images of individual SGs would improve data readability.

- In Fig. 3c, the control sample of FUS P525L seems quite different from corresponding data in Fig. 3b. Is there any difference between two experimental conditions

Reviewer #3 (Remarks to the Author):

The authors have adequately addressed my comments.

Point-to-point response to reviewers

Reviewer #1 (Remarks to the Author):

The authors have adequately addressed my earlier comments.

One item the authors might wish to consider is whether the CLSTN1 mRNA is enriching in P-bodies rather than stress granules. I suggest this possibility since the mRNA puncta in the smFISH tend to dock to the edges of stress granules, which is often where P-bodies would reside.

We thank the reviewer. Along his suggestion, we quantified the extent of localization of CLNST1 mRNA to the edges of SG. In particular, by using FIJI tools we applied a mask to select border around each granule and then segmented the smFISH signals inside the thickness of the selection. As shown in the bar plot below (N=3), the percentage of CLNST1 smFISH signals localizing to the edges of SG is around 10% in each condition, suggesting a strong tendency of CLNST1 mRNA puncta to associate to the SG core as marked by G3BP1.

We replaced the single focal planes captions of mFISH for CLSTN1 transcript (red) in fig. 2B with a more representative one.

Reviewer #2 (Remarks to the Author):

The authors addressed several comments previously raised, but I still have several remaining concerns as elaborated below. Importantly, it is still unclear whether and how altered transcriptomes are linked to SG assembly and disassembly. The manuscript describes these two phenotypic behaviors upon the knockdown or inhibition of the m⁶A writer, but how are they related? The paper still feels phenomenological, and probing this connection would significantly increase its impact.

- The authors concluded that FUS P525L SG tends to exclude m⁶A-containing transcripts. If the fractions of m⁶A-containing transcripts are compared between SG-enriched vs depleted population in FUS P525L cells, I suspect that the m⁶A-containing fraction would still be higher for SG-enriched one?

This might be because the length of transcript, which also tends to be correlated with the number of m⁶A sites, is a strong predictor of SG partitioning. But this analysis would potentially put a question mark on the author's conclusion.

The reviewer's observation is correct, and we have checked along the paper to have correctly commented this point, better explaining the observation that FUS^{P525L} SG have a lower content of m⁶A transcripts with respect to FUS^{WT} SG.

Moreover, it is true that the length contributes both to SG inclusion and to the possibility of a transcript to contain m⁶A. This information is also stated in the text:

“When analyzing the commonly enriched RNAs in FUS^{P525L} SG, we found that m⁶A-containing RNAs accounted for 22% while those of invariant species corresponded to 11% (Fig. 1E). Such difference is mainly due to an increase in the length of the RNA (Fig. S1T), further confirming previous observations that long transcripts are enriched in SG and are more likely to contain m⁶A^{17,20}. Instead, the percentage of m⁶A-containing RNAs increased up to 37% when looking at the relocated RNAs (Fig. 1E). Importantly, differently from the previous RNA subset, this enrichment is not dependent on the differential length of the transcripts since it resulted comparable to that of other transcripts included in the granules (Fig. S1T)”.

In the figure below the fraction of m⁶A containing transcripts is shown for the group of RNAs that are: 1) depleted from SG, 2) equally distributed in cytoplasm or SG, 3) enriched in SG for either FUS^{WT} or FUS^{P525L} SK-N-BE cells (bar plots). While the m⁶A-containing fraction is similar for the first two groups when comparing FUS^{WT} and FUS^{P525L}, it varies from 18% in FUS^{P525L} to 26% in FUS^{WT} for the species enriched in SG. This variation is not accompanied by a variation in transcripts length (box plots). The heatmap in the bottom describes the median value of logFC (SG /INP) for each group. These data confirm that the difference in the fraction of m⁶A-containing transcripts is not due to the length of the transcripts. We added this information in the text.

- Also, what is the m⁶A characteristics of the transcripts in the “depleted” group, those originally enriched in FUS P525L SG but depleted after METTL3 knockdown? Do they exhibit an opposite trend from the "relocated" group, like a low fraction of m⁶A-containing transcripts?

The ΔMETTL3-depleted group includes only 26 transcripts, therefore too few to perform a reliable comparison with the relocated group which includes 461 RNAs. By any means, we have analyzed all the transcripts enriched in FUS^{P525L} SG in control or METTL3 knockdown

condition and found that the more RNAs are relocated in FUS^{P525L} SG upon METTL3 knockdown, the more are methylated.

In figure below RNAs have been stratified in five equally sized groups basing of the differential enrichment in control or Δ METTL3 condition ($\Delta\log_2FC(CTRL-\Delta$ METTL3), heatmap): from left to right groups are ordered from those with higher specific enrichment in control (Δ METTL3-depleted) to those with higher specific enrichment in Δ METTL3 condition (relocated). As you can see, the m^6A -containing RNAs fraction increases together the increase of the enrichment in the Δ METTL3 condition, confirming that Δ METTL3-depleted group have an opposite trend from the Δ METTL3-enriched (relocated) group. Notably each group analyzed displayed comparable length (box plots).

- The overexpression of METTL3 increased the number of SGs even in FUS WT cells. Does it involve a change in the localization of FUS to the cytoplasm? Do these SGs show other phenotypes, such as FUS-only aggregates in recovery and reduced internal dynamics, similar to those in FUS P525L cells?

We thank the reviewer for raising this interesting point. As shown in the figure below FUS^{WT} did not change its localization upon METTL3 increase or ALKBH5 knockdown, thus FUS-only aggregates are not present in such conditions. We added this information in the text.

- The number of SGs can be affected by the expression levels of SG components (for example, overexpression of G3BP1 causes SG assembly in the absence of stress conditions; 10.1083/jcb.200502088). It is also possible that the dissolution takes longer simply because of the higher number of SGs assembled initially. Monitoring the full dissolution kinetics and the presence of irreversible species would be a better way of probing the recovery dynamics of SGs.

We thank the reviewer for this observation. Even if reported that overexpression of G3BP1 can induce SG formation in the absence of stress, in our cellular systems we observe SG formation only upon stress induction. We have added this information in the text.

Regarding the dissolution kinetics, we already investigated the physical properties of the SG upon m⁶A diminishment by using spot-variation fluorescence correlation spectroscopy (FCS) demonstrating that while the S_{conf} value for G3BP1 did not change in any condition (Fig. S4H, Fig. S4I), we observed a lower confinement (higher S_{conf} value) for FUS^{P525L} in both stress and recovery when METTL3 was inhibited (Fig. S4F, Fig. S4G). In addition, METTL3 inhibition produced a significant increase of the diffusion coefficient both for G3BP1 and FUS^{P525L} into SG (Fig. 4E, Fig. S4J), indicating that when these proteins are present in SG they assume a faster diffusion rate when m⁶A is reduced. Altogether, these data indicate that not only the SG number is changing across the conditions considered, but also their physical properties and that these alterations are relieved upon m⁶A reduction.

- Transcripts were classified into four groups based on differential enrichment in SG. My understanding is that “delMETTL3-depleted” group corresponds to transcripts originally enriched in SG but then depleted upon METTL3 knockdown. But, in smFISH data (line 199-202), GAPDH was classified into “depleted” and CALM1 into “invariant”. Based on the definition and RNA localization data, shouldn't both of them correspond to “invariant”?

We thank the reviewer for the comment, he is right. We corrected the information in the text, defining GAPDH as invariant.

- For smFISH data, adding several zoomed-in images of individual SGs would improve data readability.

Thanks to the reviewer for the useful suggestion. We added zoomed-in images of individual SGs (new fig. 2B and S2I).

- In Fig. 3c, the control sample of FUS P525L seems quite different from corresponding data in Fig. 3b. Is there any difference between two experimental conditions.

Thank you for this observation. However, aware of the fact that these experiments are intrinsically affected by biological variations, we conducted parallel stress/recovery treatments for each SK-N-BE cell line indicated, in each replicate of fig. 3c or 3b. Therefore, is not surprising to observe variations from different experiments; the important is that all the measurements are performed in parallel on the same batch of cells. The deposited raw data certify the measurements performed.

Reviewer #3 (Remarks to the Author):

The authors have adequately addressed my comments.

We thank the reviewer for his work.